# Heme controls the structural rearrangement of its sensor protein mediating the hemolytic bacterial survival

Megumi Nishinaga[1], Hiroshi Sugimoto [1,2], Yudai Nishitani [1], Seina Nagai[1], Satoru Nagatoishi [3], Norifumi Muraki[4], Takehiko Tosha [1,2], Kouhei Tsumoto[3,5,6], Shigetoshi Aono [4], Yoshitsugu Shiro [1✉] & Hitomi Sawai [1,2✉]

Hemes (iron-porphyrins) are critical for biological processes in all organisms. Hemolytic bacteria survive by acquiring $b$-type heme from hemoglobin in red blood cells from their animal hosts. These bacteria avoid the cytotoxicity of excess heme during hemolysis by expressing heme-responsive sensor proteins that act as transcriptional factors to regulate the heme efflux system in response to the cellular heme concentration. Here, the underlying regulatory mechanisms were investigated using crystallographic, spectroscopic, and biochemical studies to understand the structural basis of the heme-responsive sensor protein PefR from *Streptococcus agalactiae*, a causative agent of neonatal life-threatening infections. Structural comparison of heme-free PefR, its complex with a target DNA, and heme-bound PefR revealed that unique heme coordination controls a >20 Å structural rearrangement of the DNA binding domains to dissociate PefR from the target DNA. We also found heme-bound PefR stably binds exogenous ligands, including carbon monoxide, a by-product of the heme degradation reaction.

[1] Graduate School of Life Science, University of Hyogo, Ako, Hyogo, Japan. [2] RIKEN SPring-8 Center, Sayo, Hyogo, Japan. [3] The Institute of Medical Science, The University of Tokyo, Minato-ku, Tokyo, Japan. [4] Institute of Molecular Science, National Institute of Natural Sciences, Okazaki, Aichi, Japan. [5] Department of Bioengineering, School of Engineering, The University of Tokyo, Minato-ku, Tokyo, Japan. [6] Department of Chemistry and Biotechnology, School of Engineering, The University of Tokyo, Minato-ku, Tokyo, Japan. ✉email: yshiro@sci.u-hyogo.ac.jp; sawai@sci.u-hyogo.ac.jp

Iron is essential for the survival of all organisms and facilitates crucial cellular processes, such as respiration and DNA biosynthesis. Ferric iron is insoluble and thus inaccessible in organisms, whereas ferrous iron is soluble and is a major source of reactive oxygen species. Organisms have evolved special systems for acquiring iron and iron compounds. For example, in some pathogens, *b*-type heme (iron–protoporphyrin IX complex, from now heme means heme *b*) is a major source of iron due to the lack of heme biosynthetic genes[1–3]. These pathogens have evolved special pathways for acquiring heme from animal host blood. This causes hemolysis, which is the destruction of red blood cells, and the release of hemoglobin into the blood plasma[4–10]. Over a billion molecules of heme are released from each red blood cell in animal hosts during hemolysis. These heme molecules are both directly utilized as prosthetic groups for hemoproteins and catabolized by heme oxygenase-like enzymes to obtain iron to provide "fuel" in the cells[11,12]. Most pathogenic bacteria require $\sim 10^{-6}$ M iron to support growth, which is $\sim 10^{12}$-fold more than is available in humans[13]. However, excess heme accumulated in bacterial cells is highly cytotoxic due to reactive oxygen species generated from ferrous heme iron, which accelerate membrane peroxidation and damage to cellular proteins and DNA[14]. Cellular heme homeostasis is thus tightly regulated and bacteria have evolved sophisticated regulatory systems[4–12]. In most of these regulatory systems, the cellular heme level is monitored by heme-responsive sensor proteins: a high heme concentration engages the heme efflux system to maintain cellular heme homeostasis[15–17]. The Cys–Pro (CP) motif, in which the thiolate of Cys coordinates the heme iron as an axial ligand and the Pro residue stabilizes the Fe–Cys coordination, found in many heme-responsive sensor proteins[18,19], but several heme-responsive proteins have been recently identified that use non-CP motifs as heme-binding sites, such as the Cys–His motif in the $Ca^{2+}$-sensitive large-conductance $K^+$ (BK) channel[20], voltage-dependent $K^+$ (Kv1.4) channel[21], ATP-dependent $K^+$ ($K_{ATP}$) channel[22], and the His–His motif in HrtR (heme-regulated transporter regulator)[23,24].

In the present study, we conducted structural and functional analyses of the heme-responsive sensor protein PefR (*p*orphyrin-regulated *e*fflux *r*egulator) from the β-hemolytic bacterium *Streptococcus agalactiae*, which is the most common cause of life-threatening invasive bacterial infections (e.g., septicemia, pneumonia, and meningitis) during the neonatal period[25,26]. Fernandez et al. reported that *S. agalactiae* PefR regulates the gene expression of two newly characterized operons, *pefAB* and *pefRCD*, which are involved in heme tolerance and homeostasis in *S. agalactiae*[27,28]. *pefA* and *pefB* were predicted to encode a drug: $H^+$ antiporter and a functionally unknown protein, respectively, and *pefC* and *pefD* encode subunits of the ABC (*A*TP-*b*inding *c*assette) multidrug exporter. These operons have been suggested to work as the heme efflux system of *S. agalactiae* to prevent intracellular heme toxicity during heme acquisition. Therefore, PefR promotes the expression of these heme efflux systems in *S. agalactiae* in response to the intracellular heme concentration, likely by binding heme.

Here, we report the X-ray crystal structures of PefR in five states: apo (heme-free) PefR and its DNA complex, and holo (heme-bound) PefR and its complexes with exogenous ligands [carbon monoxide (CO) and cyanide ion ($CN^-$)]. Based on these structural data and biochemical and spectroscopic characterizations, we propose the structural basis for heme sensing by PefR, including its DNA recognition mechanism and structural changes upon heme coordination. Furthermore, we also propose that after dissociating from the target DNA, the heme of holo-PefR can stably bind CO, which is a by-product of heme degradation by heme oxygenase. These findings on a key protein required for the survival of the neonatal infection-causing hemolytic bacterium *S. agalactiae* might provide new leads for the development of antimicrobial agents effective against globally distributed drug-resistant pathogens.

## Results and discussion

**Apo-PefR and its complex with target DNA as a transcriptional regulator.** Recombinant PefR was purified in its apo form. Gel filtration analyses showed that apo-PefR eluted as a single peak with an estimated molecular mass of ~35 kDa, indicating that PefR (ca. 17 kDa in monomer) exists as a homodimer in solution (Supplementary Fig. 1). Because two putative operons (*pef*AB and *pef*RCD) are regulated by *S. agalactiae* PefR[27,28] (Supplementary Fig. 2), we examined the interaction of purified PefR protein with these operator regions by electrophoretic mobility shift assays (EMSAs; Fig. 1A). The addition of target DNA shifted the band from the position of apo-PefR protein, whereas nontarget DNA had no effect, as shown in Fig. 1A, indicating specific binding of apo-PefR for formation of its DNA complex. The equilibrium dissociation constant ($K_d$) for the apo-PefR–target DNA complex was estimated to be 40 nM by fluorescence polarization assays (Fig. 1B), indicating that apo-PefR exhibits moderately high DNA-binding affinity.

The mechanism of DNA binding to PefR was revealed by determining the crystal structures of apo-PefR and its DNA complex at 2.6 and 2.5 Å resolutions, respectively, as illustrated in Figs. 2 and 3A, and Supplementary Figs. 3, 4A, and 5, and Table 1. Both apo-PefR and its DNA complex exist as homodimers. The monomer consists of six α-helices and two β-strands containing a winged helix-turn-helix (wHTH) motif. This architecture of PefR is similar to that of the *m*ultiple *a*ntibiotic *r*esistance regulator (MarR) family of transcriptional regulators[29,30], in which helices α2–α4 and strands β1–β2 form the DNA-binding domains. Indeed, in the apo-PefR–DNA complex structure, the α3–α4 helices of the wHTH regions of both subunits interact with the major grooves; in particular, the N-terminus of the α4 helix inserts into the major groove, and the wing (the loop between β1 and β2) inserts into the flanking AT-rich minor grooves of the target DNA (Fig. 3A and Supplementary Fig. 5). Lys63 and Ser64 in the α4 helix, and Arg89 in the wing, form hydrogen bonds with DNA bases (O atoms of G8, G18, and T6, respectively; Fig. 3A, B and Supplementary Fig. 5). Lys53 (in a loop on the edge of α3), Ser62 (in a loop between α3 and α4), Ser67 (α4), Arg72 (α4), and Lys75 (α4) apparently help stabilize the complex through hydrogen-bonding or electrostatic interactions with the phosphate backbones (Fig. 3A, B and Supplementary Fig. 5). We examined the importance of these interactions for DNA recognition by conducting EMSA for several variants of PefR, in which residues interacting with DNA (Lys53, Ser62, Lys63, Ser67, Arg72, Lys75, and Arg89) were replaced with Ala (Fig. 3C). Arg89 is well conserved among proteins belonging to the MarR family, whereas the other amino acids are not conserved (Supplementary Fig. 6). A 1:1 ratio of PefR dimer to target DNA (5 μM) resulted in no band shift for the R89A variant, indicating the loss of DNA-binding activity. Under the same conditions, the K53A, S62A, K63A, S67A, and K75A variants showed activity similar to that of wild-type PefR, and the R72A variant did not bind completely with the target DNA. These EMSA results suggested that the interaction between Arg89 and T6 in the minor groove is essential for target DNA association. Although individual amino acid residues in the α3–α4 helices are not critical for sequence-specific recognition of the target DNA, they would act in synergy and contribute to DNA binding by PefR.

This crystallographic study showed that the structural characteristics of PefR are the same as those of the MarR

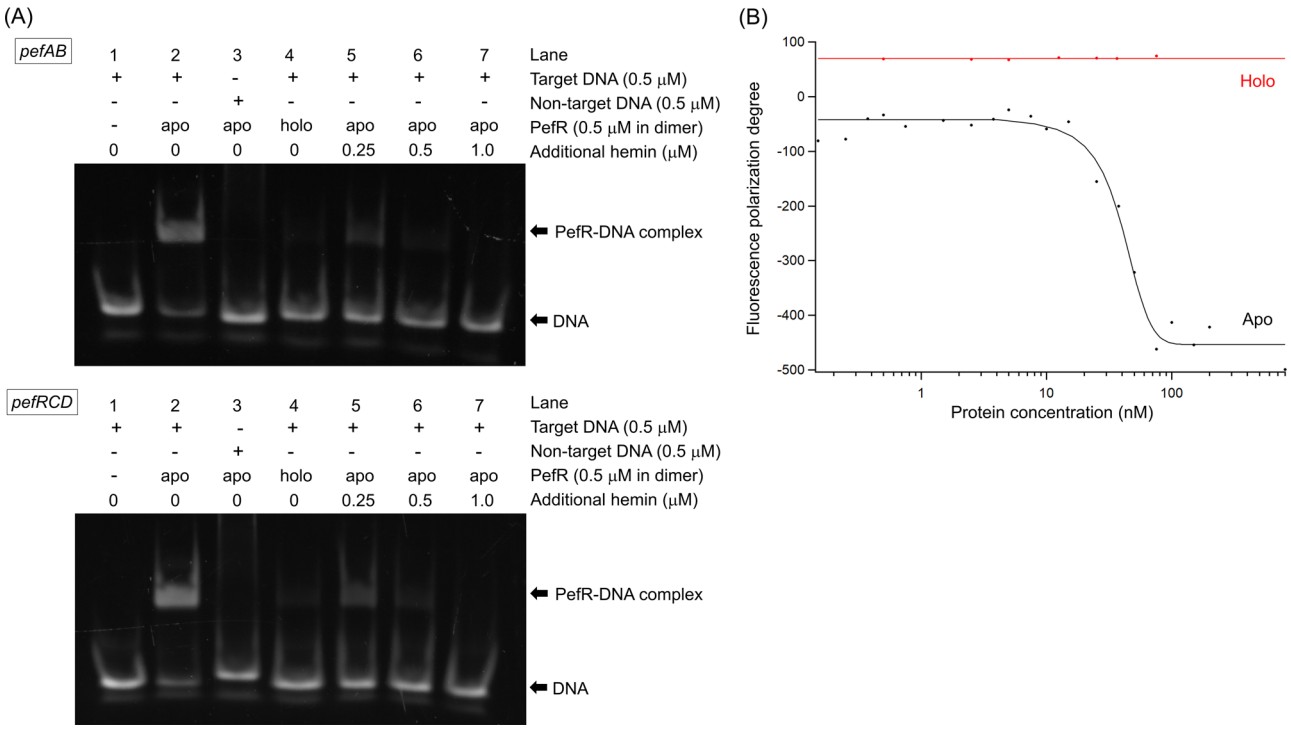

**Fig. 1 DNA-binding properties of PefR. A** EMSA assays of DNA binding with/without hemin addition. The target DNAs were designed for the *pefAB* (top panel) and *pefRCD* (bottom panel) operons. The samples were prepared by mixing PefR protein with double-stranded DNA solution [20 mM Tris-HCl (pH 7.4), 50 mM KCl, 5% glycerol, and 0.2 M MgCl$_2$], after which hemin was added to the mixture (lanes 5–7). **B** Fluorescence polarization assays with apo-PefR (black) and holo-PefR (red). The black line for apo-PefR is the calculated curve, giving a $K_d$ of 40 nM, assuming a 1:1 binding ratio of the complex of apo-PefR dimer and one double-stranded DNA.

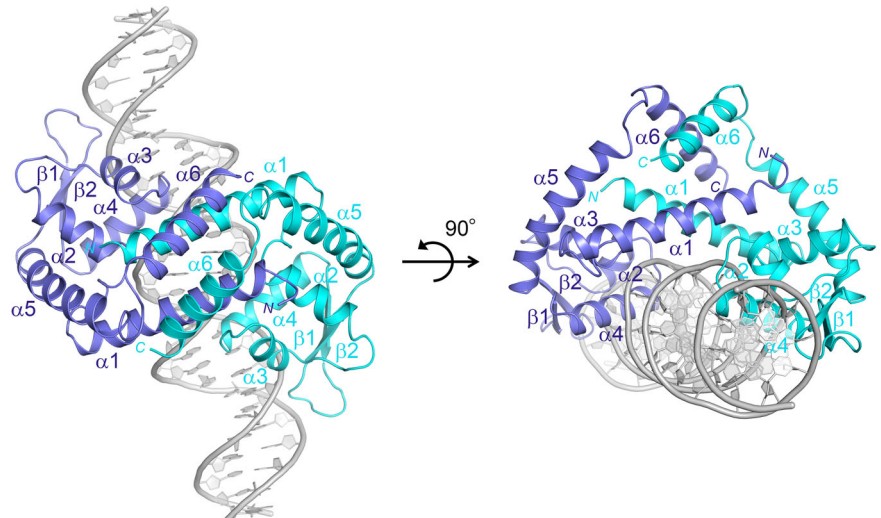

**Fig. 2 Crystal structure of the apo-PefR–DNA complex.** Overall structure of apo-PefR in the homodimer is represented by a ribbon model using blue and cyan for each subunit. The double-stranded DNA is shown in gray.

transcriptional regulator family, which regulate bacterial responses to multidrug resistance[31], oxidative stress[32], catabolism of aromatic xenobiotics[33], expression of virulence determinants[30], or transport of transition metals across the cell membrane[34]. PefR is assigned as the only *heme*-responsive protein in the family identified to date. The closest MarR structural homolog of PefR (identified using the DALI server[35]) is MepR, a multiple antibiotic resistance regulator. The structure of the apo-PefR–DNA complex is closest to that of the MepR–DNA complex (PDB code 4LLN), and the protein structure of holo-PefR (see second

paragraph of next section) is closest to that of the MepR Q18P mutant (PDB code 4LD5). DNA binding in the MepR Q18P mutant was significantly affected by creating kinks in the middle of the α1 helix and changing the orientation of each DNA-binding domain in a protomer of the dimer. Since the α1 helices form the dimer interface in MepR, as well as PefR, the Q18P mutation in MepR affects the dimerization and the distance between the DNA-binding domain of each subunit of the dimer increases, as observed in holo-PefR (vide infra). In wild-type MepR, the wHTH motif in the DNA-binding domain interacts

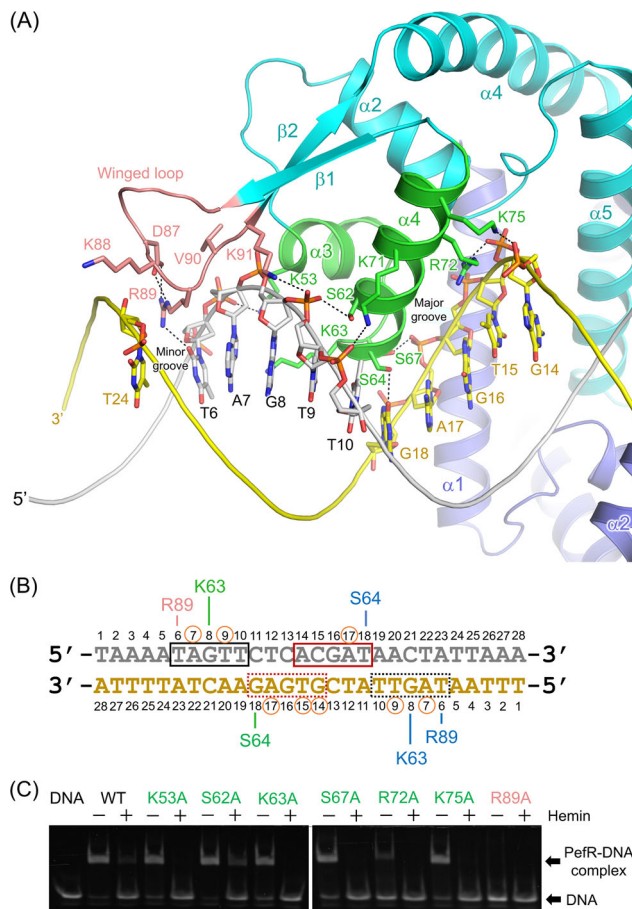

**Fig. 3 Interactions with the target DNA. A** Close-up view of the wHTH region (pink and green) and the target DNA. Hydrogen-bonding interactions are represented by dotted lines. **B** DNA sequence of the target DNA in the crystal structure. Gray- and yellow-colored letters with numbers, as well as **A**, show the bases and their sequence number in the chains of the target DNA. The black- and red-colored rectangles enclose the corresponding bases for DNA recognition by the other subunit of the PefR dimer. Amino acids with bars indicate interactions with bases, which are shown in the same color in **A**. Orange circles show positions of interactions between phosphate backbones and the side chains of amino acids. **C** EMSA of wild type and variants in the HTH region (green) and the wing loop (pink) of PefR. DNA (5 μM), hemin (10 μM) and wild-type or variant apo-PefR protein (5 μM in the dimer) were mixed for electrophoresis. Hemin addition is shown as − (without) and + (with). The target DNA was 28 bp long and contained the operator sequence for the *pefRCD* operon, which is the same sequence as used for obtaining the crystal structure.

with the consecutive major and minor grooves of the GTTAG signature sequence, and the wing loop inserts into the minor groove[36]. This DNA recognition mechanism is similar to that of PefR, in spite of the different signature sequence. In contrast, the Q18P mutation decreases the DNA-binding affinity of MepR ~2000-fold[37], similar to the DNA-binding affinity of holo-PefR (vide infra), suggesting high structural similarity between holo-PefR and the apo-MepR Q18P mutant. However, the substrate recognition mechanism of MepR remains unclear because the structural and mutational studies used the substrate-analog ethidium bromide-bound form[37].

**Heme-binding properties of PefR as a heme-responsive sensor.**
Essentially no DNA band shift was observed in the presence of holo-PefR by EMSA and fluorescence polarization (Fig. 1),

indicating no DNA binding. Furthermore, the addition of 1 equivalent of heme (two heme molecules) relative to the dimer caused full dissociation of PefR from the target DNA. These observations unambiguously indicate that PefR is a very sensitive heme-responsive sensor, in which the heme acts as an effector molecule to regulate the DNA-binding activity of PefR, and thus repressing the *pefAB* and *pefRCD* operons in *S. agalactiae*. We revealed the heme-responsive regulatory function of PefR by titrating heme into apo-PefR and following spectro-photometrically, as shown in Fig. 4A. Apo-PefR bound 1 mol equivalent of ferric heme (protein:heme = 1:1), meaning the holo-PefR binds two hemes per dimer. The dissociation constant ($K_d$) of heme from PefR was determined to be 610 nM using isothermal titration calorimetry (ITC; Fig. 4B). The binding ratio of PefR and heme based on the ITC profile was estimated to be 0.9, showing that ~1 mol equivalent of heme binds per mol equivalent of apo-PefR protein. The optical absorption spectrum of holo-PefR in the ferric heme state gave Soret and visible bands at 407 and ~530 nm, respectively, and these peaks shifted to 419 and 548 nm upon the addition of potassium cyanide (Fig. 4C). Reduction of holo-PefR shifted the Soret, α, and β peaks to 425, 529, and 559 nm, respectively. The Soret peak of ferrous holo-PefR had a shoulder ~435 nm. CO binding to ferrous holo-PefR provided peaks at 420, 537, and 569 nm (Fig. 4C).

The crystal structure of holo-PefR was determined at 1.7 Å resolution (Fig. 5, Table 1, and Supplementary Figs. 4B and 7B). The overall dimer structure of holo-PefR is an approximate isosceles triangle with a pseudo twofold symmetry axis. The α2, α5, and α6 helices and the α1 helix from the other subunit of the homodimer compose a hydrophobic heme pocket in the cavity. The heme is surrounded by the hydrophobic residues Pro4 and Leu 5 in the α1 helix; Leu15, Leu19, and Ala23 in the α2 helix; Leu31 and Ala32 in the loop between the α2 and α3 helices; Phe106, Leu110, and Met117 in the α5 helix; and Ile126 in the α6 helix (Fig. 5B). In contrast, in the apo-PefR–DNA complex, there are direct hydrophobic interactions between these residues and there is no space to accommodate a heme (Fig. 5B). Another difference in the structures of holo-PefR and the apo-PefR–DNA complex in the heme binding vicinity is a salt bridge in the apo-PefR–DNA complex between His37 in the α3 helix and Glu2 on the edge of the α1 helix of the other subunit in the dimer (Fig. 5B). Heme binding in the hydrophobic pocket formed by the hydrophobic amino acid side chains changes the orientation of each subunit in the dimer, with heme coordination altering the interaction between the α1 helices from each subunit (Supplementary Fig. 8). Several polar interactions form between Arg9 and Asn13 or Glu16 in the other subunit of the dimer upon heme binding. This mechanism is different from previously reported dimerization mechanism observed in the other heme-binding proteins, such as progesterone receptor membrane-associated component 1 (PGRMC1), in which heme–heme stacking helps dimerization of the protein[38]. In the holo-PefR structure, the heme is embedded in a solvent accessible cavity and is coordinated to His114 and the N-terminal nitrogen atom on the main chain of Met1 from the other subunit of the homodimer (Fig. 5B, and Supplementary Figs. 4B, 7B, and 8). The heme axial ligand mutant (H114A) of PefR cannot bind heme and does not dissociate from the target DNA. The $K_d$ of H114A for DNA binding is 25 nM, similar to that of wild-type apo-PefR (Supplementary Fig. 9), clearly showing that His114 imidazole binding to the heme iron acts as a trigger for structural rearrangement. Given that the side chain of His114 faces the solvent region in the apo-PefR–DNA complex (Fig. 5B), free heme molecules might easily bind to His114, with His114 thus locking heme coordination.

PefR has been reported to date from only two species, *S. agalactiae* and *Streptococcus pyogenes*, and the amino acid

**Table 1 Data collection and refinement statistics (molecular replacement).**

| | Apo-PefR (heme-free) | Apo-PefR and DNA complex | Holo-PefR | CO-bound PefR | CN⁻-bound PefR |
|---|---|---|---|---|---|
| **Data collection** | | | | | |
| Space group | $C2$ | $P3_212$ | $C2$ | $C2$ | $C2$ |
| Cell dimensions | | | | | |
| $a, b, c$ (Å) | 122.98, 46.02, 135.22 | 100.97, 100.97, 128.99 | 68.90, 32.15, 76.65 | 68.20, 31.73, 76.61 | 69.38, 31.39, 77.40 |
| $\alpha, \beta, \gamma$ (°) | 90.00, 92.81, 90.00 | 90.00, 90.00, 90.00 | 90.00, 95.07, 90.00 | 90.00, 94.34, 90.00 | 90.00, 96.80, 90.00 |
| Resolution (Å) | 50–2.6 (2.67–2.60)[a] | 40–2.5 (2.65–2.50)[a] | 30–1.7 (1.81–1.70)[a] | 40–2.1 (2.22–2.10)[a] | 40–2.1 (2.22–2.10)[a] |
| $R_{meas}$ | 3.8 (65.3) | 8.9 (191.1) | 5.4 (103.0) | 5.4 (74.9) | 5.8 (83.6) |
| $I/\sigma I$ | 18.9 (2.1)[a] | 14.6 (1.0)[a] | 16.3 (1.5)[a] | 11.7 (1.5)[a] | 14.4 (1.6)[a] |
| Completeness (%) | 97.7 (97.3)[a] | 99.9 (99.4)[a] | 97.1 (95.5)[a] | 99.8 (99.5)[a] | 99.7 (98.7)[a] |
| Redundancy | 3.7 (3.8)[a] | 7.0 (7.1)[a] | 6.6 (7.0)[a] | 3.5 (3.5)[a] | 6.3 (6.5)[a] |
| **Refinement** | | | | | |
| Resolution (Å) | 50–2.6 | 40–2.5 | 30–1.7 | 40–2.1 | 40–2.1 |
| No. reflections | 23158 | 25129 | 17175 | 9289 | 9530 |
| $R_{work}/R_{free}$ (%) | 23.6/28.8 | 21.8/24.6 | 23.7/28.5 | 22.6/28.7 | 22.7/27.7 |
| No. atoms | | | | | |
| Protein | 3704 | 2267 | 1214 | 1187 | 1172 |
| Ligand/ion | 0 | 1148 | 44 | 45 | 45 |
| Water | 0 | 0 | 37 | 24 | 17 |
| $B$-factors | | | | | |
| Protein | 107.1 | 91.3 | 53.5 | 72.9 | 68.9 |
| Ligand/ion | — | 76.2 | 65.2 | 76.4 | 73.7 |
| Water | — | — | 55.0 | 64.7 | 59.8 |
| R.m.s. deviations | | | | | |
| Bond lengths (Å) | 0.007 | 0.010 | 0.017 | 0.012 | 0.008 |
| Bond angles (°) | 0.885 | 1.89 | 1.95 | 1.89 | 1.46 |

[a]Values in parentheses are for highest-resolution shell.

sequences share 43% identity and 74% similarity. However, the sequence alignment shows a less amino acid similarity in the helix containing axial His ligand (Supplementary Fig. 10). Our heme-binding results for *S. agalactiae* PefR differ from those reported for *S. pyogenes* PefR by Sachla et al.[39]. The optical absorption spectrum of *S. pyogenes* PefR in the holo (ferric) form gave Soret bands at 435 and visible bands at 530 and 560 nm. These differences can be explained by variations in the sample preparation and the heme environment structure; *S. pyogenes* PefR had an N-terminal His tag, whereas our *S. agalactiae* PefR did not. Our crystallographic data show that the N-terminal region is important for heme binding, and thus the results of heme binding to *S. pyogenes* PefR are an anomaly. We observed structural distortions by the N-terminal His tag by comparing the X-ray crystal structures of apo-PefR with and without the N-terminal His tag (Supplementary Fig. 3). Alternatively, it is interesting that although the two PefRs share amino acid sequences with 43% identity, amino acid residues in the α5 helix, including His114 (a heme axial residue in *S. agalactiae*), are not well conserved in these PefRs (Supplementary Fig. 10).

The heme-binding motif of PefR is rare compared with that of other heme-responsive sensor proteins (see "Introduction"). His/α-amino group (αNH₂ in the N-terminal of the protein) binding from the other subunit of a dimer has been reported only for the heme-based CO sensor protein CooA from *Carboxydothermus hydrogenoformans*[40]. Heme coordination with the N-terminal amino acid residue is also found in the crystal structure of the cytochrome *f* subunit of the cytochrome $b_6f$ complex from *Brassica camoestris* (turnip)[41]. The heme in cytochrome *f* coordinates to the N-terminal Tyr1 and His25 in the same subunit. In PefR, the hydrophobic heme pocket, together with the axial ligation of His114 and the nitrogen atom on the main chain of Met1 (Fig. 5B, and Supplementary Figs. 4B, 7B, and 8), are responsible for its heme-binding affinity ($K_d$ 610 nM), which is similar to the ferric heme-binding affinity of apo-myoglobin ($K_d$ 600 nM)[42]. The $K_d$ values of other

His-coordinated heme-responsive sensor proteins for heme are lower ($K_d$ 20–120 nM)[43].

**Conformational change of PefR induced by heme binding.** Our structural comparison of the apo-PefR–DNA complex and holo-PefR sheds light on the heme-sensing mechanism. Structural events occur around the heme upon binding to PefR: the first event is coordination of each His114 from one subunit. After that three events occurs, (i) coordination of the N-terminal amino group of the other subunit, (ii) formation of a hydrophobic pocket comprising Pro4, Leu15, Ala32, Phe106, Leu110, Met117, and Leu118, and (iii) structural rearrangement of the α1 helices and the DNA-binding domains in the dimer. Accommodation of the heme should promote formation of a hydrophobic pocket in PefR (Fig. 5B and Supplementary Fig. 7), and the thus-stabilized heme will eventually sterically push the α1 helix (Supplementary Fig. 8). Rigid-body motion of the α1 helix in association with heme accommodation alters the relative orientation of the DNA-binding domain in holo-PefR from the apo form (Fig. 6A and Supplementary Movie 1), resulting in a conformational change in the DNA-binding domain (the α2, α3, and α4 helices, and the β1 and β2 strands in the wHTH region). The distance between the Cα atom of Lys63 on the edge of the α4 helix (recognition helix for interaction with the major groove of DNA) and Arg89 in the wing loop of each subunit increased from 31 to 52 Å and from 54 to 76 Å, respectively (Fig. 6B and Supplementary Movie 2). Given that the distance between two consecutive major grooves in B-DNA is ca. 34 Å, the increase in the interhelical distance between the recognition helices upon heme binding can promote the dissociation of holo-PefR from the target DNA. Our structural and functional characterization results for PefR provide the first insights into the substrate recognition and signal transduction mechanisms in a heme-responsive member of the MarR family.

The structural rearrangement observed in PefR upon heme binding causes dissociation from DNA and can be compared with

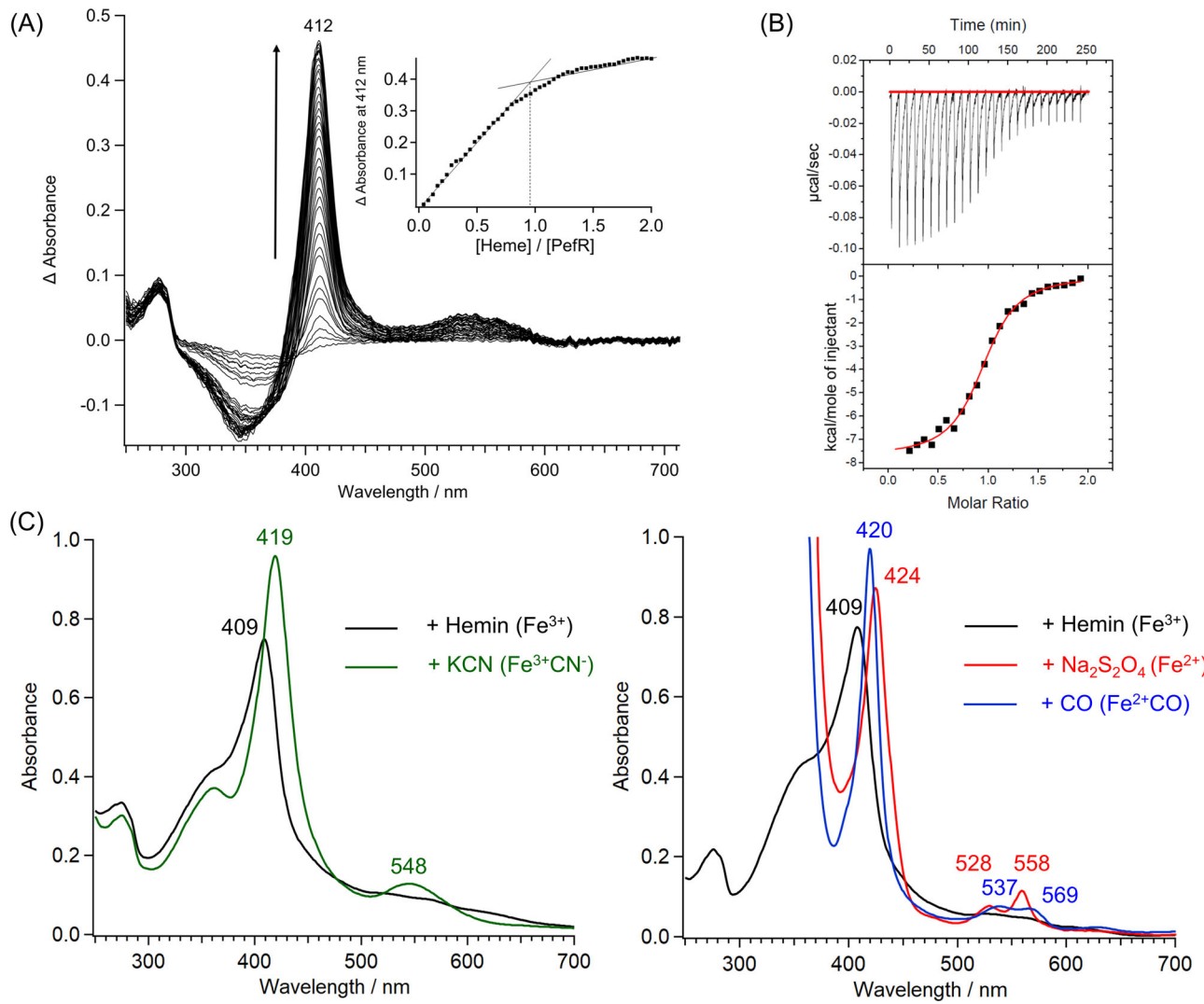

**Fig. 4 Heme-binding properties of PefR. A** Spectral change upon ferric hemin addition. Inset, titration curve of hemin binding to apo-PefR measured at 412 nm. A dimer of PefR binds two hemes, and the stoichiometry is 1:1 when considering a monomer PefR. **B** ITC of heme binding to apo-PefR. Top panel, differential heating power versus time. Lower panel, integrated and normalized heat of reaction versus the molar ratio. Experimental data are shown by black squares. Red lines show the best fit to the binding isotherm using a one-site binding model. **C** Optical absorption spectra of PefR (5 μM) in $Fe^{3+}$, $Fe^{3+}CN^-$ (left panel), and $Fe^{3+}$, $Fe^{2+}$, and $Fe^{2+}CO$ (right panel) in 40 mM Tris/HCl (pH 7.4) and 500 mM NaCl.

the rearrangements of other bacterial heme-responsive sensor proteins. Three water-soluble heme-responsive transcriptional regulators[15,23,24,27] have been reported to date: HrtR from the food and commensal bacterium *Lactococcus lactis*[23,24], HatR (heme *a*ctivated *t*ransporter *r*egulator) from the anaerobic infectious diarrhea-causing bacterium *Clostridium difficile*[15], and PefR from *Streptococci*[27]. PefR belongs to the MarR family, whereas HrtR and HatR are categorized into the *tet*racycline *r*esistance *r*egulator (TetR) family. Although structural information regarding HatR is limited, the crystal structure of HrtR[24] has been reported. In HrtR, a coil-to-helix transition upon heme binding changes the angle between the heme-anchored helix and the α1 helix in the HTH domain, moving the DNA-binding domain. The conformational changes triggered are completely different in PefR and HrtR, but in both cases the conformational changes at the heme-binding site result in changes in the center-to-center distances between the recognition helices in the DNA-binding motifs of each subunit.

**Characterization of external ligand binding to holo-PefR**. We found that holo-PefR remains stable after dissociation from target

DNA upon heme sensing. The heme-binding affinity of PefR ($K_d$ 610 nM) is similar to that of myoglobin ($K_d$ 600 nM), suggesting that heme dissociates from holo-PefR infrequently and that holo-PefR might act as a usual hemoprotein in the cytosol of *S. agalactiae*. Indeed, our spectral measurements (Fig. 4C) showed that ferric holo-PefR can combine with the external ligand $CN^-$ and that the ferrous form can combine with CO. These observations are not surprising for typical hemoproteins, but are unprecedented for heme-responsive sensor proteins. We therefore characterized ligand binding to holo-PefR using crystallographic and biochemical techniques.

We determined the crystal structures of holo-PefR with the exogenous ligands CO and $CN^-$, and found that the external ligands replace $αNH_2$ coordination, slightly shifting the N-terminal loop regions (Fig. 7, and Supplementary Figs. 4C, D and 7C, D, and Table 1). However, the overall structures are identical to that of ligand-free holo-PefR (Fig. 7A); that is, the conformation of the α1 helix is unchanged. This suggest that exogenous ligand-bound holo-PefR would have the same functional characteristics as holo-PefR, that is, no association with the target DNA. Indeed, EMSA assay of $CN^-$-bound PefR showed

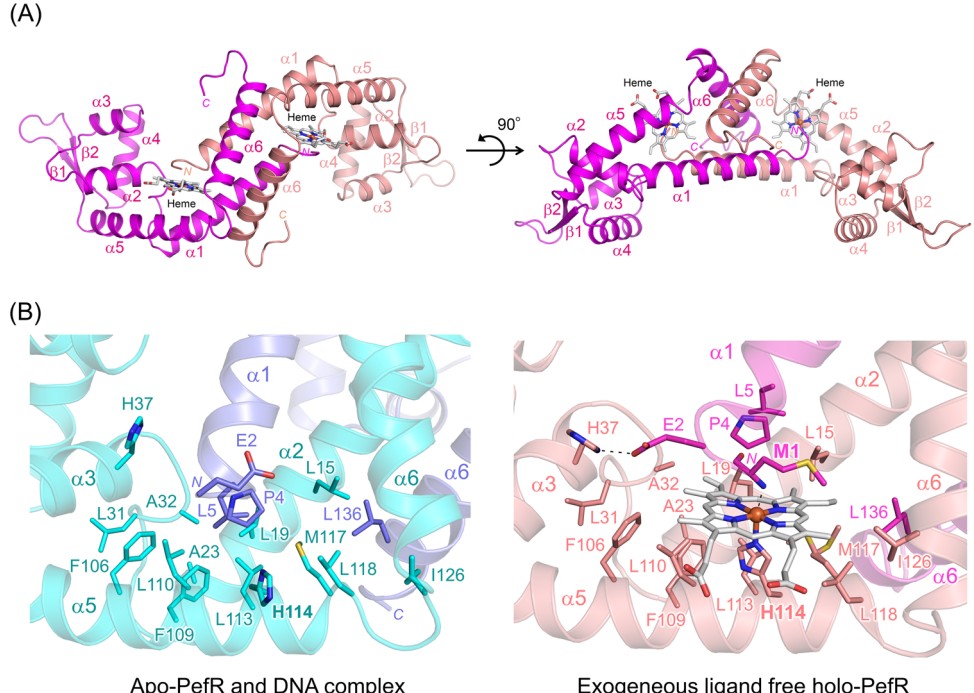

**Fig. 5 Crystal structures of holo-PefR. A** Overall structures of exogenous ligand-free holo-PefR are showed in magenta and pink. **B** Structures of the heme-binding sites in the apo-PefR–DNA complex (left panel), and in exogenous ligand-free holo-PefR (right panel). Hydrophobic residues and heme axial ligands are represented by sticks.

that the complex dissociated from the target DNA at a molar ratio of 1:1 (Supplementary Fig. 11), as observed for ferric holo-PefR. These crystallographic and biochemical findings are important for understanding heme sensing by PefR: specifically, that the N-terminal nitrogen atom of Met1 does not coordinate to the heme iron in the CO- and CN⁻-bound forms of PefR, but can dissociate from the target DNA, as is the case with holo-PefR. On the other hand, the coordination of His114 to the heme iron, and/or the formation of a hydrophobic heme pocket, might be a trigger for the structural change of the α1 helix, altering the orientation of the DNA-binding domains. In other words, heme sensing by PefR can be triggered and initially driven by the coordination of His114, and not by the terminal $\alpha NH_2$ coordinating to the heme iron.

We further investigated ligand binding to holo-PefR by determining its CO binding affinity ($K_d$) from the association ($k_{on}$) and dissociation ($k_{off}$) rate constants using flash photolysis. The CO rebinding time course of CO-bound holo-PefR was monitored at 420 nm (Supplementary Fig. 12) and the data were fitted with a double exponential. This two-phase CO association reaction would be due to the re-coordination of the N-terminal ligand following CO photodissociation. The rate constants ($k_{obs}$) at CO concentrations ranging from 0.2 to 1.0 mM were calculated from the fitted curves and were plotted against CO concentration (Supplementary Fig. 12B). Both phases corresponded to bimolecular reactions, and the calculated $k_{on}$ values were 0.13 and 0.41 $\mu M^{-1} s^{-1}$. The $k_{off}$ value of PefR was determined to be 0.014 $s^{-1}$ based on the time course measurement of CO dissociation in the presence of the oxidant potassium ferricyanide, which irreversibly oxidizes five-coordinated ferrous heme iron (Supplementary Fig. 12C). The $K_d$ values for the two-phase reaction of CO binding to holo-PefR were thus calculated as 0.11 and 0.033 µM, respectively. Comparison of the affinity and rate constants of CO binding for PefR and other heme-binding proteins (Supplementary Table 1) shows that the CO association/dissociation rate constant for PefR are similar to those for myoglobin, a typical model hemoprotein.

The results of this study provide the first structure of PefR in the CO-bound form. The CO binding properties of PefR suggest that holo-PefR can form a stable CO complex that does not associate with DNA in bacterial cells. The biological and physiological significance of these findings remain unclear, but it is notable that CO is biologically produced in cells by heme degradation systems. Heme oxygenases in host cells act as CO generators[44–46]. More than 5% of hemoglobin in host blood is saturated with CO during hemolysis, meaning that the CO concentration in the host's blood is ~115 µM (refs. [47,48]). Because the $K_d$ values for the two-phase reaction of CO binding to holo-PefR are 0.11 and 0.033 µM, PefR might strongly bind CO entering the bacterial cell from the host blood. Some bacteria possess heme degradation enzymes and generate CO through catalytic reactions[12]. Heme oxygenases in group B Streptococci, including S. agalactiae, have not been reported to date, but HupZ or a homolog heme degradation enzyme might be present in group A Streptococci and release CO through catalytic reaction[49]. Regardless, if CO is present in the cytoplasm of a bacterium, it can form a stable complex with PefR with high affinity. The CO affinity of holo-PefR is similar to that of sperm whale myoglobin, but much lower than that of some heme-based CO sensor proteins (CooA[50], RcoM-2 (ref. [51]), SUR2A[52], and cystathionine β-synthase[53,54]; Supplementary Table 1), implying that PefR does not act as a CO sensor protein. In sperm whale myoglobin, the heme-bound CO interacts with the distal His64 (ref. [55]). In holo-PefR, Pro4 is located within 3 Å of the bound CO and can interact with the CO by electrostatic or steric interactions. The residues at the heme distal side in PefR may stabilize CO binding to the heme iron, leading to tight binding of CO similar to that of sperm whale myoglobin.

**Conclusion and future perspective.** Drug-resistant infections and related morbidity and mortality are increasing globally. The World Health Organization warns of 10 million deaths and an

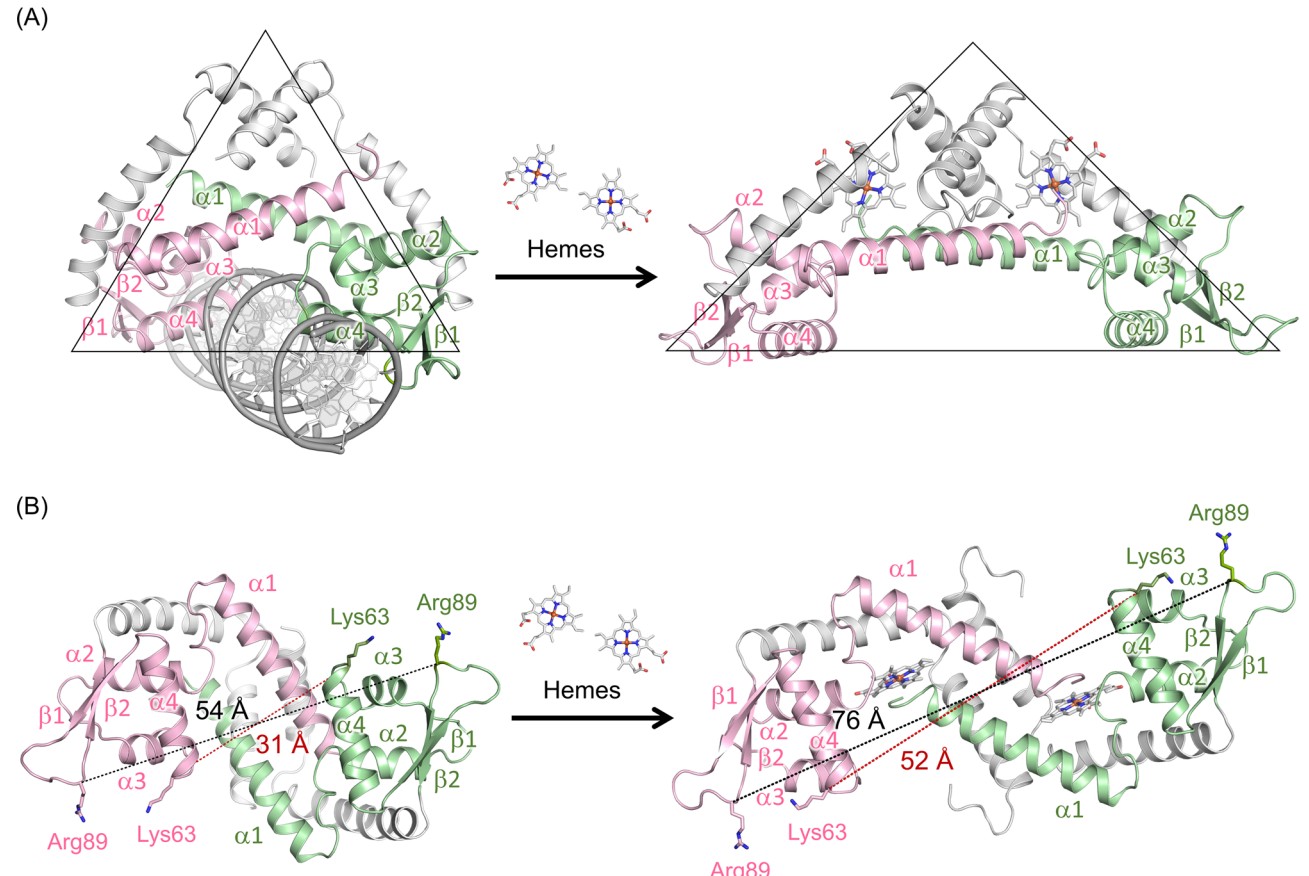

**Fig. 6 Structural changes in the conversion of apo-PefR–DNA complex (left column structures) to holo-PefR (right column structures) by heme binding.** Pink or green-colored α1 helices and the DNA-binding domains (α2–α4 helices and β1–β2 strands) highlight the location in each subunit of the homodimer structures. **A** Triangles indicate the rough shape of each structure. **B** Views 90° about the horizontal axis from **A**. The distance between the Cα atom of Lys63 on the edge of the DNA recognition helix (α4) between each subunit is shown in red dotted lines and labels, and that of Arg89 on the wing loop of the DNA-binding domain is shown in black dotted lines and labels.

economic loss of 100 trillion US dollars annually if more effective interventions against antimicrobial resistance are not applied by 2050, and identified antimicrobial resistance as one of the three greatest threats to human health[56,57]. The development of new strategies to fight bacterial infections is thus of paramount importance and many pharmaceutical companies worldwide are seeking new antibiotic lead compounds. Globally, only 21 drugs were approved as antibiotics between 2010 and 2019 (ref. [58]), but the number of drug-resistant bacteria is increasing daily. Studies on target proteins involved in the host–pathogen competition for heme provide a unique strategy to addressing this challenge. *S. agalactiae* is the cause of neonatal life-threatening invasive bacterial infections (e.g., septicemia, pneumonia, and meningitis) and does not synthesize heme[59], but relies instead on the infected hosts' blood as its major source of heme. *S. agalactiae* causes hemolysis to acquire heme as an iron source. Each host red blood cell contains over one billion molecules of heme. Heme is cyto-toxic at high concentrations, and thus excess heme released by hemolysis and entering bacterial cells must be dealt with fol-lowing infection. The detoxification of excess heme in pathogenic bacteria is a promising target for the creation of novel ther-apeutics. Here, we proposed a new concept for developing anti-microbial agents based on the heme-responsive sensor protein PefR in *S. agalactiae*. PefR is essential to the bacterial heme acquisition system and potentially enhances bacterial survival, proliferation, and infection during the hemolysis of host blood. The present study revealed that heme sensing (binding) causes

the formation of a hydrophobic heme pocket, changing the orientation of the DNA-binding domains, leading to loss of DNA-binding ability. Heme-bound holo-PefR can bind CO (Fig. 8). Inhibiting the heme detoxification mechanism of pathogens by utilizing these PefR functions could be an important strategy for developing antimicrobial agents to address the global emergency caused by drug-resistant pathogens.

## Methods

**Expression and purification of PefR.** The *S. agalactiae* *pefR* gene conjugated with HRV3C protease cleavage and C-terminal His-tag was synthesized by Eurofin Genomics, and inserted between the NdeI and BamHI sites of pET-22b(+) (Novagen) to construct an expression vector for PefR. *S. agalactiae* PefR protein fused with a N-terminal 6×His tag used only for the crystallization of apo-PefR protein was produced from pET-15b (Novagen) expression vector inserted the *pefR* gene between NcoI and BamHI sites. The codons in the synthesized *pefR* gene were optimized for protein expression in *Escherichia coli*. Site-directed mutagenesis was performed using PrimeSTAR GXL DNA polymerase (Takara Bio). Introduced mutations were confirmed by DNA sequencing by Eurofin Genomics.

*E. coli* BL21(DE3) (Nippon Gene) was used as a host for heterologous expression of PefR protein. The transformed *E. coli* cells were inoculated in 12 mL of terrific broth (TB) medium containing 50 µg⁻¹ mL⁻¹ ampicillin (Wako) and 1% glucose and cultivated for 4 h at 37 °C with shaking at 150 r.p.m. Preculture solution (1 mL) was inoculated into 300 mL of TB medium containing 50 µg⁻¹ mL⁻¹ ampicillin and the cells were cultivated at 37 °C with shaking at 120 r.p.m. for 4 h. PefR expression was induced with 0.25 mM isopropyl-β-D-thiogalactopyranoside (Protein Ark) and cultivation was continued for another 15 h at 20 °C with shaking at 80 r.p.m. The *E. coli* cells were harvested by centrifugation at 4000 × *g* for 10 min and washed with 30 mM Tris/HCl (pH 7.4). PefR was purified at 4 °C. The harvested cells were resuspended in lysis buffer [50 mM Tris/HCl (pH 7.4), 500 mM NaCl, and one tablet of cOmplete EDTA-free protease inhibitor cocktail (Roche)]. The lysate was

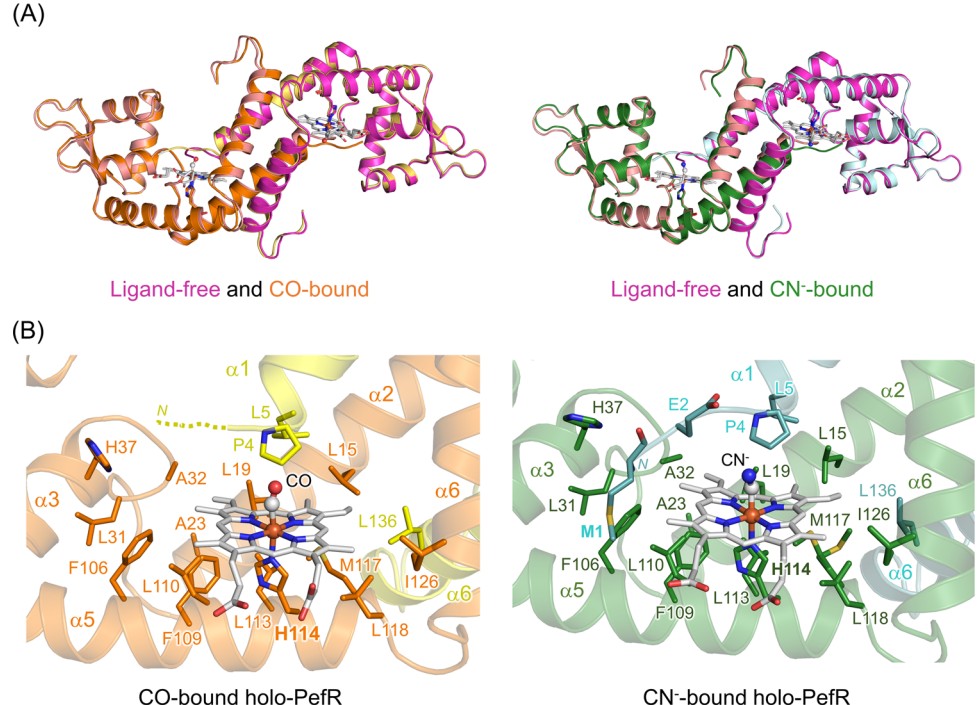

**Fig. 7 Properties of ligand-bound holo-PefR. A** Superposition of the overall structures of ligand-free holo-PefR (both panels, magenta and pink) with CO-bound holo-PefR (left panel, orange and yellow) and CN⁻-bound holo-PefR (right panel, green and cyan). **B** Heme-binding site of CO-bound (left panel) and CN⁻-bound (right panel) holo-PefRs. Porphyrin ring and the central iron atom of the heme are shown by white sticks and a red sphere, respectively. Hydrophobic residues and heme axial ligands are represented by sticks, and CO is shown in white (C atom) and red (O atom) and CN⁻ is shown in white (C atom) and blue (N atom) spheres.

**Fig. 8 Schematic diagram of the structure and function of PefR in *S. agalactiae*.** A hemolytic bacterial cell is shown in a black-outlined square. Outside area of the square shows body of infected host. Hemolytic bacteria survive by acquiring hemes from hemoglobin in red blood cells from their animal hosts. To avoid the cytotoxicity of excess heme during hemolysis, PefR act as transcriptional factors to regulate the heme efflux system in response to the cellular heme concentration. Based on crystallographic, spectroscopic, and biochemical studies, the heme coordination to DNA-bound PefR controls structural rearrangement of the DNA-binding domains to dissociate PefR from the target DNA. After dissociating from the target DNA, the heme of holo-PefR can stably bind CO, which is a by-product of heme degradation by heme oxygenase.

mixed with 0.1 mg$^{-1}$ mL$^{-1}$ lysozyme (Sigma-Aldrich), 0.05 mg$^{-1}$ mL$^{-1}$ deoxyribonuclease I (Sigma-Aldrich), and 5 mM MgCl$_2$ for 30 min and disrupted using a Microfluidizer M-110Y (Microfluidics). Cell debris was removed by ultracentrifugation at 40 k.r.p.m. for 1 h. The supernatant was loaded onto a HisTrap HP (GE Healthcare) column equilibrated with buffer A [50 mM Tris/HCl (pH 7.4), 500 mM NaCl, and 10 mM imidazole/HCl (pH 7.4)], and PefR was eluted using an imidazole concentration gradient (0 to 500 mM). For non-His-tagged PefR, the eluted fractions were treated with N-terminal hexa-His-tagged HRV3C protease (produced in-house) to remove the hexa-His tag from the recombinant PefR protein, and the solution was dialyzed against buffer B [40 mM Tris/HCl (pH 7.4) and 500 mM NaCl]. The dialyzed protein solution was loaded onto a HisTrap FF (GE healthcare) column equilibrated with buffer B to remove HRV3C protease and remaining His-tagged protein, and the flow-through was collected and concentrated using an Amicon Ultra-15 (Merck Millipore) unit with a 10 kDa cutoff by centrifuging at 15 k.r.p.m. for 20 min. The supernatant was loaded onto a HiLoad 16/600 Superdex 200 (GE healthcare) gel filtration column equilibrated with buffer C [40 mM Tris/HCl (pH 7.4), 500 mM NaCl, and 10% (w/v) glycerol]. The purity of PefR was checked by SDS–PAGE by mixing with 2× SDS–PAGE buffer containing 125 mM Tris/HCl (pH 6.8), 4% SDS, 20% (w/v) sucrose, 0.01% (w/v) bromophenol blue, and 10% (v/v) 2-mercaptoethanol and boiling at 95 °C for 10 min prior to electrophoresis on a 10% NuPAGE Bis-Tris gel (Thermo Fisher Scientific) using NuPAGE MES SDS running buffer (Thermo Fisher Scientific). The gel was stained with EzStain AQua (ATTO). The PefR sample was ≥98% pure as estimated by the SDS–PAGE. The concentration of purified PefR protein was determined based on absorption at 280 nm, where the protein concentration is 1 g$^{-1}$ L$^{-1}$ when $A_{280\,nm}$ = 0.503.

**Electrophoretic mobility shift assays**. EMSAs were performed with blunt-ended double-stranded DNA on an 8% native polyacrylamide gel using Tris borate-EDTA running buffer. The sequences of the sense strands for the target and nontarget DNA used in EMSAs were 5′- GCTATTATAAAATAGTTCTCACGA-TAACTAAAATAATG-3′ for the *pefAB* operon and 5′- TAAAA-TAGTTCTCACGATAACTATTAAA-3′ for the *pefRCD* operon (underlines are pseudo-palindromic inverted repeat sequences in each DNA sequence), and the nontarget DNA was 5′-GTTTTAAAACGACCTCGTAGCGGTCGTTAAAGG-3′. The 5′ end of the antisense strands of these DNAs was labeled with 6-carboxyfluorescein. The sense and antisense strands in equimolar amounts (25 μM final concentration) were mixed in 10 mM Tris/HCl (pH 8.0) buffer containing 1 mM EDTA (pH 8.0) and 100 mM NaCl, boiled at 100 °C for 10 min, and cooled to room temperature to obtain double-stranded DNA. The DNA fragment and apo-PefR were mixed in a 10 μL reaction mixture containing 20 mM Tris-HCl (pH 7.4), 50 mM KCl, 5% glycerol, and 0.2 M MgCl$_2$, and incubated for 20 min at room temperature. For holo-PefR, hemin (Sigma-Aldrich) dissolved in dimethyl sulf-oxide (DMSO) was mixed with the sample solution after 20 min of initial incu-bation and incubated at room temperature for an additional 20 min before electrophoresis. The gels were observed with a UV transilluminator. This experi-ment was repeated with more than three independent batches of purified PefR.

**Fluorescence polarization analysis**. A 38 bp DNA fragment (sequence of the sense strand: 5′-GCTATTATAAAATAGTTCTCACGATAACTAAAATAATG-3′) was used for fluorescence polarization analysis. The 5′ end of the antisense strand was labeled with 6-carboxy-fluorescein. Double-stranded DNA was prepared as described above under "Electrophoretic mobility shift assays". DNA fragment (10 nM) and 0–1000 nM PefR were mixed in 40 mM HEPES/Na (pH 7.4), 500 mM NaCl, and 10% (w/v) glycerol and incubated at room temperature. Fluorescence polarization was measured using a HYBRID-3000E measurement system (Photo-science). The obtained data were analyzed using the nonlinear least squares method to determine the binding constant of PefR and DNA, for which a 1:1 binding model was assumed.

**Heme titrations**. Heme titrations into apo-PefR were conducted by titrating 1 μL aliquots of 1 mM hemin solution into 1 mL of 5 μM apo-PefR dimer in 40 mM Tris-HCl (pH 7.4), 500 mM NaCl, and 10% (w/v) glycerol. Hemin was initially dissolved in DMSO and diluted to 100 μM in 40 mM Tris/HCl (pH 7.4), 500 mM NaCl, and 10% (w/v) glycerol just before use, and the concentration was corrected using a pyridine hemochrome assay[60]. Samples were mixed by inversion and incubated for 5 min before measuring the electronic absorption spectra, which were recorded at room temperature with a HITACHI U-3900 spectrophotometer.

ITC measurements of PefR with hemin solution were performed at 20 °C using a VP-ITC calorimeter (MicroCal Inc.). The apo-PefR protein was dialyzed in 40 mM HEPES/Na (pH 7.4), 500 mM NaCl, and 10% (w/v) glycerol. The external buffer following dialysis was used to dilute 20 mM hemin solubilized in DMSO to prepare a 0.2 mM hemin solution used for titration. DMSO was added to the protein solution (final concentration of 1%) and the protein concentration was adjusted to 20 μM with the dialysis buffer. Prior to loading the hemin solution, the solution was filtered using a PVDF syringe filter with a 0.22 μm pore size (Merck Millipore). For the ITC measurement, 1.5 mL of the protein solution was placed into the reaction cell of the calorimeter. The protein solution was titrated with a total of 25 injections of 10 μL spaced at 7 min intervals until the protein sample was

saturated with hemin. The experimental data were fitted to a theoretical titration curve using ORIGIN software (MicroCal Inc.).

**Protein crystallization**. Apo- and holo-PefR proteins were concentrated to 10 mg$^{-1}$ mL$^{-1}$ using an Amicon Ultra-0.5 (Merck Millipore) unit. Blunt-ended DNA duplexes 18 bp (palindromic) or 28 bp (including *pefAB* or *pefRCD* operon) in length were co-crystallized with apo-PefR to obtain the apo-PefR–DNA complex. The highest-resolution X-ray diffraction data were obtained for the crystal with the blunt-ended 28 bp duplex DNA for the *pefRCD* operon. The apo-PefR dimers and the blunt end 28 bp DNA duplex (sequence of the sense strand: 5′-TAAAATAGTTCTCACGA-TAACTATTAAAT-3′, which has a pseudo-palindromic sequence as shown with underlines) were mixed at a 1:1 molar ratio before crystallization of the apo-PefR and DNA complex. The DNA sense and antisense strands were incubated at 100 °C for 10 min and cooled to room temperature to prepare the blunt-ended 28 bp DNA duplex at a final concentration of 500 μM. Initial crystallization screening was performed using commercially available sparse matrix screens, in which 0.5 μL of protein solu-tion was mixed with an equal volume of reservoir solution and equilibrated against 50 μL of reservoir solution using the sitting-drop vapor diffusion method at 20 °C for several days. Crystals of apo-PefR were obtained using 24% (w/v) polyethylene glycol 4000, 0.2 M magnesium chloride, 0.1 M sodium acetate (pH 4.6), and 16% (v/v) 1,3-butanediol. Crystals of apo-PefR complexed with DNA were obtained using 0.2 M lithium sulfate, 0.1 M MES/Na (pH 6.5), and 25% (w/v) polyethylene glycol 2000 monomethyl ether as the reservoir solution. Crystals of holo-PefR were obtained using a reservoir solution containing 0.2 M potassium chloride, 50 mM hexamine cobalt (III) chloride, 50 mM sodium MES (pH 6.5), and 35% (v/v) 2-methyl-2,4-pentanediol. Although the crystallographic asymmetric unit consists of one protomer in holo-PefR, applying crystallographic symmetry generated a homodimer. Gel filtration studies of holo-PefR also indicated the formation of dimers in solution (Supplementary Fig. 1). CO-bound holo-PefR was crystallized by first preparing CO-bound hemin solution by adding a small amount of dithionite to 10 mM hemin under anaerobic conditions. CO-bound holo-PefR protein was prepared by mixing CO-bound hemin and apo-PefR (7 mg$^{-1}$ mL$^{-1}$) at a molar ratio of 1.2:1. The visible spectrum of the mixture solution was measured to confirm formation of CO-bound holo-PefR prior to crys-tallization. CO-bound holo-PefR was crystallized at 20 °C by a batch method using small neckless glass tubes ($\phi$ 6 mm × 32 mm, Chromacol), which were placed indi-vidually in auto-sampler screw-top glass vials (4 mL, Chromacol) with a septum screw cap (GL Science). The vials contained AGELESS® (iron powder-based O$_2$ absorber, Mitsubishi Gas Chemical) to consume any residual O$_2$. The CO-ligated PefR sample (5 μL) was mixed with the crystallization solution (5 μL) using a gas-tight syringe in room glass tube inside the screw-top vial under a CO atmosphere. Crystals were obtained in 3 days using a solution containing 5% (w/v) polyethylene glycol 400, 50 mM Bis-Tris (pH 7.0), and 24% (v/v) 2-methyl-2,4-pentanediol. The CN$^-$-bound holo-PefR protein solution was prepared using the sitting-drop vapor diffusion method by adding a final concentration of 50 mM potassium cyanide to the holo-PefR solution. CN$^-$-bound holo-PefR crystals were obtained using reservoir solution containing 0.2 M ammonium sulfate, 0.1 M Tris/HCl (pH 8.5), and 12% (w/v) polyethylene glycol 8000.

**Data collection, structural determination, and refinement**. Crystals of the apo-PefR–DNA complex, holo, CO-bound holo, and CN$^-$-bound holo-PefR forms were cryo-protected for X-ray diffraction data collection by soaking the crystals in 30% glycerol-containing reservoir solution for several minutes. Crystals of the apo form did not require additional cryoprotectant. The crystals were picked up using a cryoloop and snap-frozen in liquid nitrogen. X-ray diffraction data were collected on SPring-8 BL26B2 or BL41XU (Japan), except for apo-PefR, which was collected at SPring-8 BL44XU. The data were integrated and scaled using the program HKL2000 (ref. [61]) or XDS[62], and further processed using the CCP4 package[63]. The initial phases of holo-PefR were calculated by Solve/Resolve[64] using the MAD dataset collected at wavelengths near the absorption edge of Fe (1.7380, 1.7386, and 1.7200 Å). The phases of the DNA complex data were obtained by the molecular replacement method using Phaser[65]. A partial model of apo-PefR and 24-mer DNA of MepR (PDB code 4LLN) were used as a search model. The holo form was used as a starting model for the initial model of the structure determination of CO- and CN$^-$-bound forms. The models were manually rebuilt in Coot[66] and refined, using Refmac5 (ref. [67]) and phenix.refine[68]. Ramachandran plot revealed that >94% of residues are in favored region and <1% are in outlier region for all structures. The figures were generated by PyMol[69]. The data collection and refinement statistics are summarized in Table 1.

**CO Kinetics**. The CO association rate constant ($k_{on}$) was determined by preparing 5 μM holo-PefR in a quartz cuvette with a 10-mm path length connected to a Thunberg tube and saturating with various percentages of CO (20, 40, 60, 80, and 100%) created by a gas divider (SGD-SC-5L, HORIBA STEC). The reaction mix-ture was reduced by anaerobic mixing with small amount of powdered sodium dithionite in the Thunberg tube. CO bound to the heme iron was photodissociated by illumination with the second harmonic (532 nm) of a Q-switched Nd-YAG laser (Minilite, Continuum) with a 6-ns pulse width. The absorbance change at 420 nm upon CO rebinding was recorded with a time-resolved spectroscopic system (TSP-1000M, UNISOKU) at 293 K. The curves were fitted by a double exponential

function to obtain the apparent rate constants. The concentration of CO dissolved in the saturated buffer under various percentages of CO were estimated based on the solubility of ~1.0 mM CO in water at atmospheric pressure at 293 K (ref. [70]). The apparent rate constants ($k_{obs}$) were plotted against the CO concentrations and the $k_{on}$ values were obtained from the slope following linear fitting. The CO dissociation rate constant ($k_{off}$) was measured by the oxidation reaction of CO-bound holo-PefR. The holo-PefR sample (5 μM) was prepared in a quartz cuvette with a path length of 10 mm. The cuvette was sealed with a rubber cap and the holo-PefR was reduced by the addition of sodium dithionite (ca. 5 μM). Residual oxygen in the solution was consumed by the addition of 20 μg$^{-1}$ mL$^{-1}$ catalase, 10 mM glucose and 20 μg$^{-1}$ mL$^{-1}$ glucose oxidase, after which the air in the headspace of the cuvette was replaced with CO gas to produce CO-bound holo-PefR. The addition of various concentrations of potassium ferricyanide (25, 50, 200, 400, and 1000 μM) irreversibly oxidized the heme iron in PefR after CO dissociation. CO dissociation was monitored by measuring the absorbance change at 420 nm using a UV–visible spectrophotometer (U-3900, Hitachi) until the CO ligand was completely dissociated. The apparent rate constants ($k_{app}$) were determined from the first-order decay of the absorbance at 420 nm. Double reciprocal plots of $k_{app}$ versus the concentration of potassium ferricyanide provided $1/k_{off}$ as the $Y$-intercept of the linear fitting result.

**Statistics and reproducibility**. Proteins were expressed and purified under the same condition and all experiments were conducted in replicates as indicated. All biochemical and spectroscopic experiments were evaluated at least three biological replicates with similar results. The X-ray data collection and refinement statistics were summarized in Table 1.

**Reporting summary**. Further information on research design is available in the Nature Research Reporting Summary linked to this article.

## Data availability

The atomic coordinates and structure factors for PefR (PDB IDs: 7DVR for holo-PefR, 7DVS for apo-PefR, 7DVT for CO-bound holo-PefR, 7DVU for CN$^-$-bound holo-PefR, 7DVV for apo-PefR and DNA complex) have been deposited in the PDB (http://www.wwpdb.org). Morph movies of the conformational change between holo-PefR and apo-PefR–DNA complex have been set as Supplementary Movies 1 (side view) and 2 (top view). All Source data to generate graphs have been combined in Supplementary Data 1. DNA and protein sequences of PefR, oligo primer sequences, and DNA sequencing data for the site-directed mutagenesis have been summarized in Supplementary Data 2. Any remaining information can be obtained from the corresponding author upon reasonable request.

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

## Acknowledgements

We thank the beamline staff at SPring-8 for their assistance with data collection. X-ray diffraction experiments were performed at SPring-8 with the approval of the Japan Synchrotron Radiation Research Institute (2017A2600, 2017B2575, 2018A2576, and 2019A2519) and RIKEN (20170091), and at BL44XU under the Collaborative Research Program of the Institute for Protein Research, Osaka University (Proposal No. 2014A6955, 2014B6955, and 2015A6548). Funding for this work was provided by the Japan Society for the Promotion of Science KAKENHI Grant numbers JP26220807 (to Y.S.), JP18H02396 (H. Sugimoto), JP18K05321 (H. Sawai), and a Grant-in-Aid for Scientific Research on Innovative Areas "Integrated Biometal Science: Research to Explore Dynamics of Metals in Cellular System" 19H05761 (to Y.S.), the Fumi Yama-mura Memorial Foundation for Female Natural Scientists from Chuo Mitsui Trust and Banking (to H. Sawai), Hyogo Science and Technology Association (to H. Sawai), RIKEN Pioneering Projects "Integrated Lipidology" (to H. Sawai), and "Fundamental Principles Underlying the Hierarchy of Matter" (to Y.S.). This research was partially supported by the Cooperative Research Program of the Institute for Molecular Science under Grant number 238 and Platform Project for Supporting Drug Discovery and Life Science Research (Basis for Supporting Innovative Drug Discovery and Life Science Research [BINDS]) for AMED under Grant number JP18am0101094. We are grateful to Prof. Emma. L. Raven (University of Bristol, UK) for stimulating discussions and critical reading of the manuscript.

## Author contributions

H. Sawai designed the study. S.A. and H. Sawai prepared the *E. coli* expression system for recombinant PefR protein. M.N., Y.N., S. Nagai, N.M., and H. Sawai prepared the PefR samples. M.N. and Y.N. measured the optical absorption spectra. M.N., S. Nagai, and N.M. crystallized the PefR. M.N., H. Sugimoto, S. Nagai, N.M., and H. Sawai collected and analyzed the X-ray diffraction data. M.N., S. Nagatoishi, K.T., and H. Sawai performed the ITC measurements. M.N., Y.N., and T.T. conducted the flash photolysis measurements. M.N., N.M., S.A., and H. Sawai conducted the fluorescence polarization analyses. M.N., H. Sugimoto, Y.N., N.M., T.T., S.A., Y.S., and H. Sawai wrote the manuscript. All authors analyzed the data and discussed the results.

## Competing interests

The authors declare no competing interests.
