## [Peer Review File · Communications Biology]

Reviewers' comments:

Reviewer #1 (Remarks to the Author):

In this study, Nishinaga and co-workers report on the structural characterization of the transcription regulator PefR from *S. agalactiae* in different states (apo-, DNA-bound, heme-bound, CO-heme-bound and CN-heme-bound-PefR). They also report the functional characterization of the protein showing that PefR binds DNA in its apo-form and that upon heme binding, the protein cannot bind DNA any more, suggesting a mechanism for heme-response. They conclude that, in addition to be a transcription regulator, PefR may also act as a CO scavenger to help protecting the pathogen from CO release upon heme degradation by the host. Overall the structural and functional work provides lots of valuable information that might deserve publication in *Communications Biology*. However, the reviewer would like to raise several issues in the manuscript that must be fixed before any publication. Indeed, the organization of the manuscript makes it difficult to follow and some important information is missing, despite being available from the presented data. For example, the authors start the Results section with their functional data but they need to refer to the heme-binding mode on the fifth line although the structure was not presented yet. The reviewer would suggest to reorganize the manuscript, which is clearly a structural work, starting with the structural description of PefR. Regarding the structural description, a comparison between the DNA-free apo form and the heme-bound form would be appreciated to better assess the flexibility range of the HTH-domain. Would the authors consider the observed conformations as resting ones or as ones stabilized by the crystal packing constraints?

Stereoviews of Figs 1B and 4C would be greatly appreciated to see better the interactions with DNA and heme, respectively.

The authors mention that their results are apparently different from those for another PefR, but they do not explain in what extend and what are these differences.

There is a problem when referring to MepR in page 7. PDB code 4LD5 does not correspond to holo-PefR as indicated in the manuscript.

In page 8, the authors state that "Under the current situation, our results of structural and functional characterization of PefR provide the first definitive insights into substrate recognition and its signal transduction mechanisms in a heme responsive member of MarR family." Beside the term definitive that might be a bit too strong, at that stage, the authors did not provide any clue about the mechanism for substrate recognition and signal transduction. The structural description for the heme-binding mode only occurs in the following section. Furthermore, despite they have apo-, holo- and DNA-bound structures at hands, the authors never describe in deep details their structures to extract such information. A more extensive structural analysis might probably give them more insights into how heme binds to the protein and how this binding would trigger structural changes that results in an increase in the spanning between the two HTH domains.

When describing the heme-binding mode (page 8), the authors refer to an N-terminal nitrogen atom but there is only one N-ter nitrogen atom per chains, so please correct to the N-terminal nitrogen atom.

In page 8, the authors mention a CP motif, but it is not clear for the reviewer why referring to this at that stage. Maybe mention the CP motif in the introduction to better highlight the new result from PefR structure.

The authors claim that H114 is key to bind heme. However, from the presented results, it is not clear whether this residue is accessible in the apo-form to indeed bind heme and induce the formation of the hydrophobic pocket or if the heme has to be sandwiched in the active site pocket before and that H114 would lock that binding later. This point should be further discussed.

Regarding a proposed role for PefR as a putative CO scavenger, this seems, according to the presented data, highly speculative. For instance, do the authors know the protein concentration. The reviewer asks whether the usual concentration for transcription regulators is pretty low and thus binding released CO would only happen because of the intrinsic heme reactivity toward CO and thus is not a physiological scavenging function. Additional data would certainly be required to support this assumption. Without these data, the reviewer would suggest the authors to reduce its description to a

minimum in the discussion.

Finally, the manuscript would also require a significant proof-reading by a native English-speaking colleague to remove all the grammatical errors, word missing..., to ease the reading.

Provided all these modifications are made, the reviewer would consider this manuscript as suitable for publication in Communications Biology.

Reviewer #2 (Remarks to the Author):

The manuscript presents interesting data on a heme sensor regulator, namely its structure in the heme-free and heme-bound states, and complexed with DNA, as well as with CO and CN⁻ bound hemes. It shows a rare type of heme coordination, which in itself is interesting, but not unique – for example, in cyt f from the b6f complex, the heme is bound by a histidine and the N-terminal amino group of a histidine. Regarding the structural part my main comment is that most figures are very difficult to look at, with an overload of hydrogen bonds and amino acids labelled. If not simplified, or expanded in the main ms or in the supplemental material, it will be quite hard to catch all the relevant information.

My major concern relates to the kinetics part, and the putative role of PefR as a CO scavenger. Regarding the second part, it is pure speculation. Most heme proteins bind, in the ferrous form, CO, albeit with different affinities.

Regarding the kinetics of binding and release of CO – why is CO binding biphasic, if there is only one heme? Secondly, for CO release, since the authors oxidized the protein with ferricyanide, what they are really measuring is not the rate constant for CO release, but rather a redox-linked CO release, which does not allow determining the thermodynamic equilibrium constant, as done in the ms. Also, the heme-titration does not yield 1 heme per dimer, at most about 0.8, suggesting some kind of heterogeneity in the sample.

Finally, there are a lot of details that need to be corrected, which I commented in the attached file.

Reviewer #3 (Remarks to the Author):

The manuscript written by Nishinaga et al. provided a quality of crystallographic characterization of PefR, the heme-responsive sensor protein in *Streptococcus agalactiae*. High-resolution crystallographic analyses revealed that PefR

1. The description in the line 182~184 in Page 8 showed a unique finding of this study in that the role of His/a-amino group in the N-terminal of the protein has been only reported for the heme-based CO sensor protein like CooA. In mammals, cystathionine beta-synthase has the ability to sense CO at physiological levels; in this case his residue of the N-terminus (and the cysteine residue) serves as an axial ligand for heme that allows CO binding and translocation of the cys residue at the opposite side to inhibit the enzyme activity.

2. In PefR, heme helps dimerization of PefR through a unique mechanism provided by authors. However, heme has been known to help dimerization through the heme-stacking mechanism in other proteins. It occurs only in bacterial proteins but also those in mammals (e.g. PGRMC1). The authors should briefly describe these lines of discussion which strengthen the uniqueness of the dimerization mechanisms of PefR as compared with other heme-dimerized proteins by citing previous references.

3. The authors argued that CO is cytotoxic for host defense mechanisms. This reviewer agrees that ferrous heme enhances oxidative stress. However, the products of heme oxygenase such as biliverdin and CO at physiological concentrations are cytoprotective in mammals. Furthermore, some species in bacterial kingdom utilize CO as a substrate for CO dehydrogenase to synthesize organic compounds such as acetyl CoA. The present data just indicated that PefR actually capture CO but does not degrade the gas into other metabolites. Just like a function of myoglobin in mammals, can PefR deliver CO towards other bacteria to transfer the gas as a nutritional source ? Without any experimental evidence to empower the authors' hypothesis, the authors need to soften the text indicating toxicity of CO without citing previous publication suggesting physiologic significance of the gas.

4. (Page2, line 82) ,,but also a novel function for the detoxification of excess CO which is a heme degradation product from host's blood. Can PefR effectively scavenge "toxic CO in blood" in the presence of huge amounts of Hb ? Can the authors provide any experimental evidence to prove the hypothesis, or show any discussion based on the data shown in Supplementary Table 2 ? This should be clarified in the revised version.

5. CO or CN binds to ferrous or ferric heme of PefR, respectively. These data were shown in the manuscript. CN is a nutrient for some bacterial species, while CN serves as a toxicant released from the pathogenic bacterial species. Why did not the authors mention physiological roles of CN capturing by PefR ?

6. Can the authors provide any information of mechanisms by which PefR heme is maintained ferrous? How can the protein allow CO binding in vivo?

Minor:

Page 3, line 59: fight against bacterial infections.....

Page 6, line 118: ,,while that of non-target DNA did not,,,,,,

Page 6, line 136: as DNA-binding domain

Page 7, line 145: EMSA should be shown by full-length at least once at the 1st one (electrophoretic mobility shift assay).

Page 14, line 315; delete " as"

Response to Reviewers (COMMSBIO-20-1913-T)

We divided the comments from reviewers 1 and 2 and numbered them, together with our responses, as shown below.

Reviewer #1 (Structural biology of metalloproteins):

Comment-1: In this study, Nishinaga and co-workers report on the structural characterization of the transcription regulator PefR from *S. agalactiae* in different states (apo-, DNA-bound, heme-bound, CO-heme-bound and CN-heme-bound-PefR). They also report the functional characterization of the protein showing that PefR binds DNA in its apo-form and that upon heme binding, the protein cannot bind DNA any more, suggesting a mechanism for heme-response. They conclude that, in addition to be a transcription regulator, PefR may also act as a CO scavenger to help protecting the pathogen from CO release upon heme degradation by the host. Overall the structural and functional work provides lots of valuable information that might deserve publication in *Communications Biology*. However, the reviewer would like to raise several issues in the manuscript that must be fixed before any publication.

Response to comment-1: Thank you for your comments. We have revised the manuscript accordingly.

Comment-2: Indeed, the organization of the manuscript makes it difficult to follow and some important information is missing, despite being available from the presented data. For example, the authors start the Results section with their functional data but they need to refer to the heme-binding mode on the fifth line although the structure was not presented yet. The reviewer would suggest to reorganize the manuscript, which is clearly a structural work, starting with the structural description of PefR.

Response to comment-2: The revised manuscript was reorganized, with more in-depth descriptions given in the order apo-PefR-DNA complex, holo form, and CO-bound form, when describing the molecular mechanisms of DNA dissociation from PefR upon heme sensing.

Comment-3: Regarding the structural description, a comparison between the DNA-free apo form and the heme-bound form would be appreciated to better assess the flexibility range of the HTH domain. Would the authors consider the observed conformations as resting ones or as ones stabilized by the crystal packing constraints?

Response to comment-3: Apo-PefR (DNA-free) protein was difficult to crystallize, but we succeeded by using N-terminal 6xHis tagged PefR protein. The structural flexibility of the protein was limited by the tag, which possibly aided packing in the crystal. Therefore, the structure of apo-PefR (DNA-free) is not that of the protein in the natural resting state. We show structural and flexibility differences between apo-PefR (DNA-free) and the protein part of the apo-PefR-DNA complex in Supplementary Figure 3 in the revised manuscript.

Comment-4: Stereoviews of Figs 1B and 4C would be greatly appreciated to see better the interactions with DNA and heme, respectively.

Response to comment-4: Figures showing the interactions with DNA and heme were redrawn and

added as Supplementary Figures 5 and 7.

Comment-5: The authors mention that their results are apparently different from those for another PefR, but they do not explain in what extend and what are these differences.

Response to comment-5: To date, PefR has been characterized in only two species (*S. agalactiae* and *S. pyogenes*). Differences in their amino acid sequences and optical absorption spectra are described on lines 180-190 of pages 8-9.

Comment-6: There is a problem when referring to MepR in page 7. PDB code 4LD5 does not correspond to holo-PefR as indicated in the manuscript.

Response to comment-6: The MepR Q18P mutant (PDB code 4LD5) shows very low DNA binding due to kinks in the middle of the $\alpha 1$ helices, which change the orientation of each protomer of the dimer. Because holo-PefR cannot bind to DNA, these structures might be similar. We added new explanatory text on lines 126-133 on pages 6-7.

Comment-7: In page 8, the authors state that “*Under the current situation, our results of structural and functional characterization of PefR provide the first definitive insights into substrate recognition and its signal transduction mechanisms in a heme responsive member of MarR family.*” Beside the term *definitive* that might be a bit too strong, at that stage, the authors did not provide any clue about the mechanism for substrate recognition and signal transduction.

Response to comment-7: The word “definitive” has been removed from the sentence. Because this sentence summarizes this work, this sentence was transferred to lines 219-221 in page 10.

Comment-8: The structural description for the heme-binding mode only occurs in the following section. Furthermore, despite they have apo-, holo- and DNA-bound structures at hands, the authors never describe in deep details their structures to extract such information. A more extensive structural analysis might probably give them more insights into how heme binds to the protein and how this binding would trigger structural changes that results in an increase in the spanning between the two HTH domains.

Response to comment-8: Thank you for this advice. We provide insights into how heme binds to the protein and how this binding would trigger structural changes based on our structures of PefR in the newly added section “Conformational change of PefR induced by heme binding” from line 203, page 9 as well as in morphing movies (Supplementary Movies 1 and 2).

Comment-9: When describing the heme-binding mode (page 8), the authors refer to an N-terminal nitrogen atom but there is only one N-ter nitrogen atom per chains, so please correct to the N-terminal nitrogen atom.

Response to comment-9: Thank you for this advice. “An N-terminal nitrogen atom” has been changed to “the N-terminal nitrogen atom” in the revised manuscript.

Comment-10: In page 8, the authors mention a CP motif, but it is not clear for the reviewer why referring to this at that stage. Maybe mention the CP motif in the introduction to better highlight the new result from

PefR structure.

Response to comment-10: Sentences explaining the CP motif were transferred to the Introduction section (lines 54-58, page 3) to help describe the properties of heme-responsive sensor proteins.

Comment-11: The authors claim that H114 is key to bind heme. However, from the presented results, it is not clear whether this residue is accessible in the apo-form to indeed bind heme and induce the formation of the hydrophobic pocket or if the heme has to be sandwiched in the active site pocket before and that H114 would lock that binding later. This point should be further discussed.

Response to comment-11: His114 is essential for binding heme, judging by the properties of the H114A mutant (Supplementary Figure 9). We suggest that the order of structural changes upon heme binding to apo-PefR-DNA complex are i) coordination of His114 from one subunit, ii) coordination of the N-terminal amino group of the other subunit, iii) formation of a hydrophobic pocket comprising Pro4, Leu15, Ala32, Phe106, Leu110, Met117 and Leu118, iv) structural rearrangement of the α 1 helices and DNA binding domains in the dimer. Because the side chain of His114 faces the solvent region in the apo-PefR-DNA complex, free heme molecules might easily bind to His114. This explanatory text was added to the revised manuscript (lines 173-179, page 8 and lines 205-208, page 9).

Comment-12: Finally, the manuscript would also require a significant proof-reading by a native English-speaking colleague to remove all the grammatical errors, word missing..., to ease the reading. Provided all these modifications are made, the reviewer would consider this manuscript as suitable for publication in *Communications Biology*.

Response to comment-12: The revised manuscript has been edited by a native English speaker and a professional proofreader. We hope the revised manuscript is now suitable for publication in *Communications Biology*.

Reviewer #2 (Metalloproteins and heme binding):

Comment-1: The manuscript presents interesting data on a heme sensor regulator, namely its structure in the heme-free and heme-bound states, and complexed with DNA, as well as with CO and CN⁻ bound hemes. It shows a rare type of heme coordination, which in itself is interesting, but not unique – for example, in cyt f from the b6f complex, the heme is bound by a histidine and the N-terminal amino group of a histidine. Regarding the structural part my main comment is that most figures are very difficult to look at, with an overload of hydrogen bonds and amino acids labelled. If not simplified, or expanded in the main ms or in the supplemental material, it will quite hard to catch all the relevant information.

Response to comment-1: Thank you for your interest in the findings of our study and for your advice. Although the N-terminal ligand comes from another subunit in PefR and CooA dimers, cytochrome *f* in the *b6f* complex is coordinated to the N-terminal Tyr1 and His25 in the same subunit¹. A sentence describing heme coordination by cytochrome *f* was added to the revised manuscript (lines 194-197, page 9).

Comment-2: My major concerns relates to the kinetics part, and the putative role of PefR as a CO scavenger. Regarding the second part, is pure speculation. Most hemeproteins bind, in the ferrous form, CO, albeit with different affinities.

Response to comment-2: You are correct that most hemoproteins bind CO to ferrous heme. Because we could not obtain experimental evidence for the putative role of PefR as a CO scavenger (see Appendix, attached file), these speculative sentences have been removed from the revised manuscript.

Comment-3: Regarding the kinetics of binding and release of CO – why is CO binding biphasic, if there is only one heme? Secondly, for CO release, since the authors oxidized the protein with ferricyanide, what they are really measuring is not the rate constant for CO release, but rather a redox-linked CO release, which does not allow determining the thermodynamic equilibrium constant, as done in the manuscript.

Response to comment-3: We are currently investigating why CO binding is biphasic. The effect is likely due to rebinding of the N-terminal nitrogen atom of Met1 in the other subunit of the PefR dimer after the photodissociation of CO. It is unlikely due to the allosteric effect of the homodimer because CO binding to the heme does not induce significant conformational changes from ligand-free holo PefR. Detailed properties of CO binding will be reported in a future publication.

Comment-4: Also, the heme-titration does not yield 1 heme per dimer, at most about 0.8, suggesting some kind of heterogeneity in the sample.

Response to comment-4: We carefully performed the heme-titration experiment using a temperature controller and a magnetic stirrer in the sample cell. The new data replace the old and are shown as Fig. 4A in the revised manuscript. This result shows approx. 1 heme per protein molecule. ITC heme-titration data showed that the binding ratio of PefR and heme (N) was approx. 0.9, as analyzed by curve fitting (one-site binding model) using the program ORIGIN. This means that 1 heme molecule bound per PefR monomer, so the PefR dimer binds a total of 2 heme molecules. Heme-titration experiments always have some associated error due to protein assay, heme solubility, and the like. Thus, even if the ratio is 1:1, the data may not exactly show a 1:1 ratio.

Comment-5: Finally, there are a lot of details that need to be corrected, which I commented in the attached file.

Response to comment-5: Thank you for your careful reading of our original manuscript and your comments. Responses to comments [r1] to [r26] in the attached PDF file are given below.

Comment [r1]: Heme should be substituted by Heme B, as there are a lot of other hemes, e.g., a, b, c, d, etc. This should be changed throughout the text, or saying something like – from now one Heme B will be simply designated Heme, in the Introduction

In heme B, the protoporphyrin ix protoporphyrin IX

Response to comment [r1]: Our understanding of your comment is that you recommended that we explain the heme type acquired in bacteria. The heme used in this study is released from hemoglobin in host red blood cells. An explanation of this has been added to the ABSTRACT and the INTRODUCTION

sections (lines 23-24, page 2 and lines 40-42, page 3). Also, we revised “protoporphyrin ix” to “protoporphyrin IX” in No. 27 on the list of REFERENCES.

Comment [r2]: Heme *b* is scarcely soluble in water; therefore, it is not present as such neither in the host blood nor inside bacteria. It is always bound to some other protein. This comment also applies to several sentences in the ms

Response to comment [r2]: The solubility and cytotoxicity of heme are explained in the revised manuscript (lines 38-39, page 3).

Comment [r3]: Within the host and also the bacteria, when they also possess heme oxygenases.

Response to comment [r3]: Bacterial heme oxygenases also generate CO by heme degradation. However, IsdG and IsdI from *Staphylococcus aureus*² as well as MhuD from *Mycobacterium tuberculosis*³ are unable to generate CO by the reaction. Although HupZ is reported as a heme degradation enzyme and releases CO in Group A Streptococci, there are no such reports for Group B Streptococci, including *Streptococcus agalactiae*. The properties of HupZ from Group A Streptococcus sp. were reported based on the crystal structure and biochemical data⁴. Using the amino acid sequence of HupZ, we performed a BLAST search to identify the putative protein based on the genomic database of *S. agalactiae* and found that the amino acid sequence of a pyridoxamine 5'-phosphate oxidase family protein matched well with that of HupZ (Figure A). This protein could be a candidate heme degradation enzyme in *S. agalactiae*, releasing CO into the bacterial cell. Therefore, we removed “within the host” from the revised manuscript.

Figure A: Sequence alignment of HupZ from Group A Streptococcus sp. (labeled “SpHupZ”, upper lines) and a pyridoxamine 5'-phosphate oxidase family protein from *S. agalactiae* (labeled “Putative”, bottom lines, sequence ID of NCBI database WP_053039123.1). The amino acid sequences share 96% identity.

Comment [r4]: Since not all heme is degraded by the host heme oxygenase, the amount of CO liberated by this process may be low- Is it known?

Response to comment [r4]: In healthy males, the concentration of hemoglobin in blood is 13.1 to 16.6 g/dL, which is approximately 2.3 mM. During hemolysis of host blood, the ratio of CO-saturated hemoglobin increases to more than 5%^{5,6}, compared with 1% under normal healthy conditions⁷. If 5% of the hemoglobin is CO-saturated during hemolysis, the CO concentration in the host blood is at least 115 μ M. Given that the K_d of holo-PefR and CO is *ca.* 20 nM, PefR might strongly bind CO in host blood.

Comment [r5]: Only ferric iron is insoluble, not ferrous iron.

Response to comment [r5]: Please refer our response to comment [r2].

Comment [r6]: Many non-pathogens have also heme acquiring systems.

Response to comment [r6]: Yes, many non-pathogens have heme acquiring systems. However, we would like to focus here on pathogens, given that most are heme auxotrophs due to a lack of heme biosynthetic genes. We added an explanation of this on lines 40-44, page 3.

Comment [r7]: This is a wild speculation – to act as CO detoxifier its intracellular concentration would have to be quite high, and the amount of PefR is not known

Response to comment [r7]: The intracellular amount of PefR is unknown, but it should be upregulated in the presence of excess heme. However, we could not obtain experimental evidence for the role of PefR as a CO detoxifier and thus speculative sentences were removed from the revised manuscript.

Comment [r8]: The inset of Fig 1A shows a stoichiometry lower than 1 -about 0.8

Response to comment [r8]: Please refer to our response to Comment-4 for Reviewer-1.

Comment [r9]: Why hydrophobic?

Response to comment [r9]: His114 is not involved in the hydrophobic interaction. The interaction arises from hydrophobic residues Pro4 and Leu 5 in the α 1 helix, Leu15, Leu19 and Ala23 in the α 2 helix, Leu31 and Ala32 in the loop between the α 2 and α 3 helices, Phe106, Leu110 and Met117 in the α 5 helix and Ile126 in the α 6 helix in the heme binding pocket (lines 157-160, pages 7-8).

Comment [r10]: Which those?

Response to comment [r10]: “Those” was changed to “ K_d of other His coordinated heme-responsive sensor proteins” in the revised manuscript (line 200, page 9).

Comment [r11]: In Supplementary Fig3 is shown a chromatogram of PefR, without indication of whether is the apo or holo form. Apart from the clarification, both should be shown, and it is incorrect to use molecular weight – it is molecular mass, as here in the main text.

Response to comment [r11]: Chromatograms of both the apo and holo forms are shown in Supplementary Figure 2. The molecular mass of the PefR dimer was added to the main text (lines 83-85, page 5).

Comment [r12]: Other amino acids, I believe.

Response to comment [r12]: “Others” was changed to “individual amino acids” in the revised manuscript (line 116, page 6).

Comment [r13]: Being a homo dimer, why does it binds only one heme?

Response to comment [r13]: Two hemes bind to a homo-dimer, but not in a straightforward manner in PefR. One heme is coordinated to the N-terminal nitrogen atom of the main chain of Met1 in the A-chain and His114 in the B-chain, and the second heme is coordinated to Met1 in the B-chain and His114 in the A-chain. This is described in more detail in the revised manuscript.

Comment [r14]: To call the CP motif as a heme binding site is weird -, since the heme does not bind to either of these amino acids in PefR.

Response to comment [r14]: This paragraph was revised for clarity.

Comment [r15]: But in PefR the heme ligands are the His imidazole nitrogen and the N-terminal amino group of methionine.

Response to comment [r15]: Please see our response to comment [r14].

Comment [r16]: Why biphasic, since there is only one heme? Sample heterogeneity?

Response to comment [r16]: Please refer to the Response to Comment-3.

Comment [r17]: This is not correct at all. This would be a redox-driven CO dissociation, which can not neither be called K_{off} nor used for the calculation of K_{d} .

Response to comment [r17]: The explanatory text has been revised. The oxidant only oxidized the five-coordinated ferrous heme formed by spontaneous dissociation of CO, and this oxidation is irreversible. The dependency of K_{app} on oxidant concentration is due to the rate of oxidation after CO dissociation.

Comment [r18]: This sentence is incomplete (and in effect unnecessary)

Response to comment [r18]: This sentence was removed from the revised manuscript.

Comment [r19]: Which by the way have never been reported.

Response to comment [r19]: This description of the physiological function of CO was removed from

the revised manuscript.

Comment [r20]: See comments at the beginning of the ms.

Response to comment [r20]: Our speculation regarding this function was removed from the revised manuscript.

Comment [r21]: Confusing, needs rephrasing.

Response to comment [r21]: This sentence was rephrased in the revised manuscript.

Comment [r22]: If it were a scavenger, it had to exist in high concentrations and would not act as a transcriptional regulator.

Response to comment [r22]: The idea of a CO scavenger was removed from this paragraph.

Comment [r23]: Rephrase.

Response to comment [r23]: This sentence was rephrased.

Comment [r24]: This is absolutely not true. Just an example – *S. aureus*.

Response to comment [r24]: Yes. Not all pathogenic bacteria are heme auxotrophs. But in Group B *Streptococcus* strains, including *Streptococcus agalactiae*, the source bacterium of PefR, are also reported as heme auxotrophs⁸ and lack some or all of the enzymes needed for heme biosynthesis.

Comment [r25]: Highly repetitive, all is said already above.

Response to comment [r25]: This paragraph was revised to remove repetitive sentences.

Comment [r26]: Idem, and comments along the ms.

Response to comment [r26]: All of your comments were taken into account. Thank you for your help in improving our manuscript.

Reviewer #3 (CO sensor proteins):

Comment: The manuscript written by Nishinaga *et al.* provided a quality of crystallographic characterization of PefR, the heme-responsive sensor protein in *Streptococcus agalactiae*. High-resolution crystallographic analyses revealed that PefR

1. The description in the line 182~184 in Page 8 showed a unique finding of this study in that the role of His/ α -amino group in the N-terminal of the protein has been only reported for the heme-based CO sensor protein like CooA. In mammals, cystathionine beta-synthase has the ability to sense CO at physiological levels; in this case His residue of the N-terminus (and the cysteine residue) serves as an axial ligand for

heme that allows CO binding and translocation of the Cys residue at the opposite side to inhibit the enzyme activity.

Response to 1: Thank you for this information on cystathionine beta-synthase (CBS). The heme coordination structure of the CO-bound form of CBS (hexa-coordinated ferrous heme with axial ligands of His65 and CO)⁹ is similar to that of PefR. However, in CBS, His65 and Cys53 are coordinated to the heme iron, and Cys53 in the N-terminal heme-binding domain binds to CO in the CO-bound form, rather than the first residue from the N-terminus as is the case in PefR¹⁰. However, CBS is reported to be a heme-based CO sensor protein^{11,12}. Therefore, references to CO sensing and the binding kinetics of CBS were added to the revised manuscript in line 290, page 13.

2. In PefR, heme helps dimerization of PefR through a unique mechanism provided by authors. However, heme has been known to help dimerization through the heme-stacking mechanisms in other protein. It occurs only in bacterial proteins but also those in mammals (e.g. PGRMC1). The authors should briefly describe these lines of discussion which strengthen uniqueness of the dimerization mechanisms of PefR as compared with other heme-dimerized proteins by citing previous references.

Response to 2: Dimerization through the heme-stacking mechanism in PGRMC1 is has been added and is compared with PefR in lines 168-171, page 8.

3. The authors argued that CO is cytotoxic for host defense mechanisms. This reviewer agrees that ferrous heme enhances oxidative stress. However, the products of heme oxygenase such as biliverdin and CO at physiological concentrations are cytoprotective in mammals. Furthermore, some species in bacterial kingdom utilize CO as a substrate for CO dehydrogenase to synthesize organic compounds such as acetyl CoA. The present data just indicated that PefR actually capture CO but does not degrade the gas into other metabolites. Just like a function of myoglobin in mammals, can PefR deliver CO towards other bacteria to transfer the gas as a nutritional source? Without any experimental evidence to empower the authors' hypothesis, the authors need to soften the text indicating toxicity of CO without citing previous publication suggesting physiologic significance of the gas.

Response to 3: Thank you for this information on CO physiology in mammals and bacteria. We could not obtain direct evidence of CO delivery from PefR to other bacteria, and thus our description of CO toxicity in the revised manuscript has been shortened.

4. (Page2, line 82) ,, but also a novel function for the detoxification of excess CO which is a heme degradation product from host's blood. Can PefR effectively scavenge "toxic CO in blood" in the presence of huge amounts of Hb? Can the authors provide any experimental evidence to prove the hypothesis, or show any discussion based on the data shown in Supplementary Table 2? This should be clarified in the revised version.

Response to 4: Please read our Response to comment [r4] for Reviewer-2. We estimated that the CO concentration in host blood is approximately 115 μ M. Given that the K_d of holo-PefR and CO is *ca.* 20 nM, PefR might strongly bind CO in host blood. However, holo-PefR is present in the cytoplasm of bacterial cells. We do not know how much CO is transferred from the host's blood to the bacterial cell interior.

5. CO or CN binds to ferrous or ferric heme of PefR, respectively. These data were shown in the manuscript. CN is a nutrient for some bacterial species, while CN serves as a toxicant released from the pathogenic bacterial species. Why did not the authors mention physiological roles of CN capturing by PefR?

Response to 5: There is no report of CN utilization as a nutrient or a toxicant in *Streptococci* and thus we do not mention the physiological roles of CN capture by PefR.

6. Can the authors provide any information of mechanisms by which PefR heme is maintained ferrous? How can the protein allow CO binding in vivo?

Response to 6: *Streptococcus agalactiae* was recently reported as an electrogenic bacterium that can transfer electrons extracellularly across the cell envelope to or from electron acceptors, including electrodes, oxide minerals, and other bacteria^{13,14}. Such extracellular electron transfer allows the bacteria to acquire energy for growth and reproduction, or to facilitate communication between cells. The electrons inside bacterial cells are provided by the oxidation of NAD(P)H¹⁴. This electron transfer system could provide a possible mechanism by which PefR heme is maintained in the ferrous form in bacterial cells. The ferrous form of PefR could bind CO produced by heme oxygenase in the bacterial cells.

Minor:

Page 3, line 59: fight against bacterial infections.....

Page 6, line 118: ,,.,while that of non-target DNA did not,,,,,,

Page 6, line 136: as DNA-binding domain

Page 7, line 145: EMSA should be shown by full-length at least once at the 1st one (electrophoretic mobility shift assay).

Page 14, line 315; delete “as”

Response to Minor: Thank you for pointing out the above mistakes. All have been corrected in the revised manuscript.

REFERENCES

1. Musiani, F., Dikiy, A., Semenov, A. Y. & Ciurli, S. Structure of the Intermolecular Complex between Plastocyanin and Cytochrome f from Spinach. *J. Biol. Chem.* **280**, 18833–18841 (2005).
2. Matsui, T. *et al.* Heme Degradation by *Staphylococcus aureus* IsdG and IsdI Liberates Formaldehyde Rather Than Carbon Monoxide. *Biochemistry* **52**, 3025–3027 (2013).
3. Nambu, S., Matsui, T., Goulding, C. W., Takahashi, S. & Ikeda-Saito, M. A New Way to Degrade Heme. *J. Biol. Chem.* **288**, 10101–10109 (2013).
4. Sachla, A. J. *et al.* In vitro heme biotransformation by the HupZ enzyme from Group A streptococcus. *BioMetals* **29**, 593–609 (2016).
5. Blumenthal, I. Carbon Monoxide Poisoning. *J. R. Soc. Med.* **94**, 270–272 (2001).
6. Widdop, B. Analysis of carbon monoxide. *Ann. Clin. Biochem. Int. J. Lab. Med.* **39**, 378–391 (2002).

7. Kinoshita, H. *et al.* Carbon monoxide poisoning. *Toxicol. Reports* **7**, 169–173 (2020).
8. Joubert, L. *et al.* Visualization of the role of host heme on the virulence of the heme auxotroph *Streptococcus agalactiae*. *Sci. Rep.* **7**, 1–13 (2017).
9. Taoka, S. *et al.* Human cystathionine β -synthase is a heme sensor protein. Evidence that the redox sensor is heme and not the vicinal cysteines in the CXXC motif seen in the crystal structure of the truncated enzyme. *Biochemistry* **41**, 10454–10461 (2002).
10. Taoka, S., West, M. & Banerjee, R. Characterization of the Heme and Pyridoxal Phosphate Cofactors of Human Cystathionine β -Synthase Reveals Nonequivalent Active Sites †. *Biochemistry* **38**, 2738–2744 (1999).
11. Shintani, T. *et al.* Cystathionine β -synthase as a carbon monoxide-sensitive regulator of bile excretion. *Hepatology* **49**, 141–150 (2009).
12. Kabe, Y. *et al.* Cystathionine β -synthase and PGRMC1 as CO sensors. *Free Radic. Biol. Med.* **99**, 333–344 (2016).
13. Tahernia, M. *et al.* Characterization of Electrogenic Gut Bacteria. *ACS Omega* **5**, 29439–29446 (2020).
14. Pankratova, G., Hederstedt, L. & Gorton, L. Extracellular electron transfer features of Gram-positive bacteria. *Anal. Chim. Acta* **1076**, 32–47 (2019).

Reviewer #1 (Remarks to the Author):

In this revised version of their manuscript, Nishinaga and co-workers now present a clearer analysis of the transcription regulator PefR from *S. agalactiae* in different states (apo-, DNA-bound, heme-bound, CO-heme-bound and CN-heme-bound-PefR). The authors fulfilled most of the previous reviewer's recommendations. However, the reviewer would like to raise several remaining points that must be fixed before any publication.

Lines 131-134. The argument is rather weak because the MepR Q18P variant cannot easily be compared to the holo-PefR. Indeed, introducing a kink in the middle of an alpha-helix is somewhat different to binding a ligand. Could the authors be more accurate/specific when doing the comparison?
Line 182 PefR instead of PerR

Line 187 the authors state that the results previously published on *S. pyogenes* PefR are an anomaly. I would suggest the authors to be more cautious because their own data do not fully support this statement. Indeed, having a His-tagged protein may impair NH₂-terminal binding, but the authors results using CO and CN also suggest that such NH₂-binding is not essential, thus raising the question about its conservation in *S. pyogenes* PefR.

Lines 204-207. As already mentioned in the previous review, the proposed succession of events is speculative. At least one would expect some justification for proposing this order. Could the authors elaborate more about this?

Lines 251-260. This paragraph is not clear. If the reviewer picked it well, the authors mean that NH₂-terminal binding to the heme is not essential for triggering conformational changes responsible for DNA release. Please rephrase.

Provided all these modifications are made, the reviewer would consider this manuscript as suitable for publication in *Communications Biology*.

Reviewer #2 (Remarks to the Author):

This interesting ms was substantially improved upon revision, and I have only four minor remarks

1- 54 The Cys-Pro (CP) motif forms the heme binding site of many heme responsive sensor proteins^{18,19} but several heme-responsive proteins have been recently identified that

I still did not understand what means the "Cys-Pro" motif forms the heme binding site, if the ligands are a His and the N-terminus NH₂.

2- 84 as a single peak with an estimated molecular weight of about 35,000

Should be molecular mass and please add after 35,000 "Da"

3- 121 PefR is assigned as the only heme-responsive
122 protein in the family identified to date, reported thus far

"reported thus far" is redundant

4- 145 Apo-PefR bound 1 mol equivalent of ferric heme (protein:heme = 1:1).

Written like this is misleading- the holoprotein binds 2 hemes per dimer, and the stoichiometry is 1:1 only when considering a monomer. Here and in the respective figure captions it would be better to put mol Heme/dimer

Reviewer #3 (Remarks to the Author):

While an experiment to determine physiological role of the sensor protein, the authors fully addressed questions raised by this reviewer. Discussion was well revised on the basis of previous publications to be cited.

Response to Reviewers (COMMSBIO-20-1913A)

Reviewer #1

In this revised version of their manuscript, Nishinaga and co-workers now present a clearer analysis of the transcription regulator PefR from *S. agalactiae* in different states (apo-, DNA-bound, heme-bound, CO-heme-bound and CN-heme-bound-PefR). The authors fulfilled most of the previous reviewer's recommendations. However, the reviewer would like to raise several remaining points that must be fixed before any publication.

Comment-1 (line 131-134): The argument is rather weak because the MepR Q18P variant cannot easily be compared to the holo-PefR. Indeed, introducing a kink in the middle of an alpha-helix is somewhat different to binding a ligand. Could the authors be more accurate/specific when doing the comparison?

Response to comment-1: Based on the crystal structure of the Q18P MepR (PDB code 4LD5), the $\alpha 1$ helices including Gln18 form the dimer interface. Since the Q18P mutation in MepR affects the dimerization by the kink of the $\alpha 1$ helices, the distance between DNA binding domain of each subunit of the dimer increases, as observed in holo-PefR. This explanation was added to lines 128-131, page 6.

Comment-2 (line 182): PerR instead of PefR.

Response to comment-2: Thank you for finding our typo. We revised it.

Comment-3 (line 187): The authors state that the results previously published on *S. pyogenes* PefR are an anomaly. I would suggest the authors to be more cautious because their own data do not fully support this statement. Indeed, having a His-tagged protein may impair NH₂-terminal binding, but the authors results using CO and CN also suggest that such NH₂-binding is not essential, thus raising the question about its conservation in *S. pyogenes* PefR.

Response to comment-3: The difference between *S. pyogenes* and *S. agalactiae* PefRs can be explained by variations in the sample preparation and the heme environment structure. Since the second reason (heme proximal environment) was missing in our previous manuscript, the explanation was added to line 193-196, page 9.

Comment-4 (lines 204-207): As already mentioned the previous review, the proposed succession of events is speculative. At least one would expect some justification for proposing this order. Could the authors elaborate more about this?

Response to comment-4: Most probably, the heme coordination to the His114 is essential and the first event for the structural change. This suggestion can be supported by the H114A mutant, which weakens heme binding, and could not be dissociated from the target DNA upon

the addition of heme. It is also clear that the final event is dissociation of the heme-bound PefR from target DNA. However, as was pointed out by the reviewer, there were no direct evidence to show the order of the events; “coordination of the N-terminal amino group”, “formation of a hydrophobic pocket” and “structural rearrangement of the DNA binding domains”. Therefore, this paragraph was amended in the revised manuscript, as was shown in lines 211-214, pages 9-10.

Comment-5 (lines 251-260): This paragraph is not clear. If the reviewer picked it well, the authors mean that NH₂-terminal binding to the heme is not essential for triggering conformational changes responsible for DNA release. Please rephrase.

Response to comment-5: Thank you for your understanding about the triggering conformational changes. To clarify our appealing point for the readers, this paragraph was amended, as is shown in the revised manuscript (line 262, page 11).

Reviewer #2

This interesting ms was substantially improved upon revision, and I have only four minor remarks.

Comment-1: “54-55 The Cys-Pro (CP) motif forms the heme binding site of many heme-responsive sensor proteins but several heme-responsive proteins have been recently identified that”
I still did not understand what means the “Cys-Pro” motif forms the heme binding site, if the ligands are a His and the N-terminus NH₂.

Response to comment-1: The “Cys-Pro” motif is sometimes found in the heme-containing sensor proteins as the heme-iron ligand motif, in which the thiolate of Cys coordinates to the heme as an axial ligand, and the next Pro residue stabilizes the Fe-Cys coordination. These structural and functional characteristics are described in the papers, which are cited as ref 18, 19. In addition, an explanation was added to lines 54-55, page 3.

Comment-2: “84 as a single peak with an estimated molecular weight of about 35,000”
Should be molecular mass and please add after 35,000 “Da”.

Response to comment-2: We revised to “molecular mass” and “35 kDa” (line 85, page 5).

Comment-3: “121-122 PefR is assigned as the only heme-responsive protein in the family identified to date, reported thus far”
“reported thus far” is redundant.

Response to comment-3: In the revised manuscript, the phrase was removed (line 123, page 3).

Comment-4: “145 Apo-PefR bound 1 mole equivalent of ferric heme (protein:heme =1:1).”
Written like this is misleading- the holo protein binds 2 hemes per dimer, and the stoichiometry is

1:1 only when considering a monomer. Here and in the respective figure captions it would be better to put mol Heme/dimer.

Response to comment-4: Thank you for the advice. The moles of heme for holo-PefR was added after the sentence (lines 148-149, page 7), and also in the figure caption of Figure 4(A).

Reviewer #3

While an experiment to determine physiological role of the sensor protein, the authors fully addressed questions raised by this reviewer. Discussion was well revised on the basis of previous publications to be cited.

Response to the comment: We appreciate the time and effort required to review our manuscript. Thank you so much for your understanding.